# Hepatic SerpinA1 improves energy and glucose metabolism through regulation of preadipocyte proliferation and UCP1 expression

Shota Okagawa[1], Masaji Sakaguchi [1] ✉, Yuma Okubo[1], Yuri Takekuma[1], Motoyuki Igata[1], Tatsuya Kondo[1], Naoki Takeda[2], Kimi Araki [2,3], Bruna Brasil Brandao[4], Wei-Jun Qian [5], Yu-Hua Tseng [4], Rohit N. Kulkarni [6,7], Naoto Kubota[1], C. Ronald Kahn [4] & Eiichi Araki [1]

Lipodystrophy and obesity are associated with insulin resistance and metabolic syndrome accompanied by fat tissue dysregulation. Here, we show that serine protease inhibitor A1 (SerpinA1) expression in the liver is increased during recovery from lipodystrophy caused by the adipocyte-specific loss of insulin signaling in mice. SerpinA1 induces the proliferation of white and brown preadipocytes and increases the expression of uncoupling protein 1 (UCP1) to promote mitochondrial activation in mature white and brown adipocytes. Liver-specific SerpinA1 transgenic mice exhibit increased browning of adipose tissues, leading to increased energy expenditure, reduced adiposity and improved glucose tolerance. Conversely, SerpinA1 knockout mice exhibit decreased adipocyte mitochondrial function, impaired thermogenesis, obesity, and systemic insulin resistance. SerpinA1 forms a complex with the Eph receptor B2 and regulates its downstream signaling in adipocytes. These results demonstrate that SerpinA1 is an important hepatokine that improves obesity, energy expenditure and glucose metabolism by promoting preadipocyte proliferation and activating mitochondrial UCP1 expression in adipocytes.

Metabolic syndrome results from complex dysregulation of metabolic homeostasis in tissues including liver, adipose tissue, muscle and intestine and is associated with reduction of insulin signaling and action, i.e., insulin resistance. Obesity, which is due to varying combinations of excess energy intake and decreased energy utilization, is one of the important driving factors of various metabolic disorders, including type 2 diabetes, dyslipidemia, hypertension and coronary heart disease[1]. Obesity is also associated with a change in the balance of white adipose tissue (WAT) and brown adipose tissue (BAT)[2]. WAT stores energy as triglycerides (TGs). Excess WAT causes obesity and is

[1]Department of Metabolic Medicine, Faculty of Life Sciences, Kumamoto University, 1-1-1 Honjo, Chuoku Kumamoto, Japan. [2]Institute of Resource Development and Analysis, Kumamoto University, Kumamoto, Japan. [3]Center for Metabolic Regulation of Healthy Aging, Kumamoto University, Kumamoto, Japan. [4]Section of Integrative Physiology and Metabolism, Joslin Diabetes Center, Harvard Medical School, Boston, MA, USA. [5]Biological Sciences Division, Pacific Northwest National Laboratory, Richland, Washington, USA. [6]Section of Islet Cell & Regenerative Biology, Joslin Diabetes Center, Harvard Medical School, Boston, MA, USA. [7]Department of Medicine, BIDMC and Harvard Stem Cell Institute, Harvard Medical School, Boston, MA, USA. ✉e-mail: msakaguchi@kumamoto-u.ac.jp

associated with adipose tissue inflammation and increased secretion of leptin, fatty acid-binding protein (FABP4), and proinflammatory cytokines, such as tumor necrosis factor-α (TNF-α), interleukin-6 (IL-6) and monocyte chemoattractant protein-1 (MCP-1)[3,4].

BAT, on the other hand, is involved in thermogenesis and increased energy expenditure. This occurs through increased levels of expression of uncoupling protein-1 (UCP1), primarily stimulated by the sympathetic nervous system following cold exposure or exercise[2,5,6]. BAT consumes energy and helps maintain body temperature, ultimately protecting against obesity and inflammation of adipose tissue[2,5–7]. Upon cold exposure or exercise, WAT adipocytes can also undergo transformation into thermogenic beige adipocytes that express UCP1, similar to adipocytes in BAT but at somewhat lower levels[8,9]. However, developmentally these are different, with white and beige adipocytes being derived from myogenic factor 5 (Myf5)-negative precursor cells, while brown adipocytes arise from Myf5-positive precursor cells[8]. It has been shown that β-adrenergic receptor stimulation following cold exposure activates transcription factors such as PGC1α and PRDM16, increasing UCP1 expression and causing heat production in mitochondria[8,9]. Studies have shown that the amount of active BAT in humans is negatively correlated with body mass index (BMI)[10–12]. Large clinical surveys have shown that individuals with BAT detected by [18]F-fluorodeoxyglucose positron emission tomography–computed tomography scans have lower prevalence of metabolic syndrome phenotypes and cardiometabolic disease[13]. The presence of BAT also independently correlated with lower odds of type 2 diabetes, dyslipidemia, coronary artery disease, cerebrovascular disease, congestive heart failure and hypertension[13]. Thus activation of BAT and browning of WAT appear to protect against body fat accumulation and its comorbidities.

Interestingly, many of the complications associated with obesity, including insulin resistance, fatty liver disease and increased risk of cardiovascular disease, are also associated with extreme loss of body fat, i.e., lipodystrophy[14]. Experiment lipodystrophy created by either constitutive or inducible adipose tissue-specific knockout of insulin receptor (IR) alone (Ai-IRKO) or adipose tissue-specific double knockout of the IR and insulin-like growth factor 1 (IGF1) receptor (Ai-DKO) have demonstrated the critical roles of adipocytes and adipocyte insulin sensitivity in whole-body metabolic balance[15]. Indeed, Ai-IRKO and Ai-DKO mice exhibit a marked acute decrease in adipose tissue mass due to unsuppressed lipolysis and increased apoptosis of adipocytes within three days of induction of recombination and subsequently develop hepatosteatosis, hyperglycemia, and hyperinsulinemia with hyperproliferation of pancreatic β-cells[15]. In the absence of continuous induction of gene recombination, however, adipocytes can be recovered, and both WAT and BAT can regenerate in mice reversing the severe metabolic syndrome[15]. The regeneration of adipose tissues and reversal of metabolic syndrome are associated with robust proliferation of preadipocytes and rapid differentiation of these cells into new mature adipocytes[15]. Serum from Ai-DKO mice and conditioned culture medium from liver slices of Ai-DKO mice can stimulate preadipocyte proliferation in vitro, suggesting that soluble factors from the liver contribute to the reactivation of adipocytes and reversal of metabolic syndrome[16].

Using liquid chromatography-tandem mass spectrometry (LC-MS/MS)-based proteomics, we compared the molecules upregulated in the serum during recovery from the metabolic disease between Ai-DKO mice and control mice. Here we found that hepatic serine protease inhibitor A1 (SerpinA1) was among the most upregulated proteins in Ai-DKO mice and that SerpinA1 can indeed activate brown fat and promote browning of white fat to increase energy expenditure and improve glucose tolerance. Conversely, SerpinA1 knockout (SPA1KO) mice show adipocyte mitochondrial dysfunction, obesity, energy expenditure impairment, and hyperglycemia with systemic insulin resistance. Mechanistically, we show that SerpinA1 binds to the

EphrinB2 receptor, thus controlling p38 phosphorylation and subsequently regulating UCP1 expression. Thus SerpinA1 is an important liver-derived secretory factor (hepatokine) that regulates adipocyte function and systemic metabolism.

## Results

### SerpinA1 induces preadipocyte proliferation

Mice with an inducible knockout of the insulin receptor in adipocytes (Ai-IRKO) and inducible double knockout of both IR and IGF1 receptor (IGF1R) in adipocytes (Ai-DKO) using tamoxifen-inducible adiponectin-CreERT2 mice exhibit rapid development of lipodystrophy, with almost completely brown and white adipocyte loss, and severe metabolic syndrome15. In both of these models, normal metabolism is spontaneously restored after withdrawal of tamoxifen due to rapid preadipocyte proliferation and adipocyte regeneration driven by a yet unidentified liver-derived factor[15,16]. To begin to identify the serum factors that promote preadipocyte proliferation, we have evaluated the change in the serum proteome of Ai-DKO mice before and at 3 days after induction of lipodystrophy by tamoxifen injection (Fig. 1a). Among the 1660 quantified proteins, 67 were significantly upregulated and 68 were downregulated in serum of Ai-DKO mice compared to Controls using thresholds of $p < 0.05$ and an absolute |Log$_2$ (fold change Ai-DKO/Control)| > 0.3 (Fig. 1b). ApoC3 was the most upregulated protein in Ai-DKO mice and did show some stimulatory activity on preadipocyte proliferation in vitro, however, ApoC3 did not affect adipocyte recovery in Ai-DKO mice, indicating that other circulating molecules must be driving recovery from this syndrome[16].

Another candidate upregulated in Ai-DKO mice serum 3 days after tamoxifen injection was SerpinA1, which was increased in level by 1.5-fold (Fig. 1b). Similar to Ai-DKO mice, Ai-IRKO mice also showed adipose tissue regeneration from lipodystrophy after withdrawal of tamoxifen[15], and, like the Ai-DKO mice, also showed a 1.4- to 1.6-fold increase in serum SerpinA1 levels 3 days after tamoxifen injection (Fig. 1c). This increase was not observed in IGF1R-only inducible adipocyte-specific knockout mice (Ai-IGF1RKO). SerpinA1 is encoded by the *SERPINA1* gene, also known as α1-antitrypsin (A1AT), since it inhibits neutrophil elastase, a proteolytic enzyme produced by neutrophils during inflammation[17,18]. The *SERPINA1* gene is located on chromosome 14 in humans and chromosome 12 in mice, where it has multiple paralogs (termed SerpinA1a-e). An analysis of *SerpinA1* mRNA expression in various organs and tissues of male and female C57BL/6 mice using a primer set that detects all five murine SerpinA1 paralogs and primers sets that recognize individual SerpinA1 paralogs, respectively, revealed that all paralogs of *SerpinA1a-e* mRNA are abundantly expressed in the liver and not expressed or expressed at significantly lower levels in other organs and tissues (Fig. 1d and Supplementary Fig. 1a). We examined the effect of recombinant SerpinA1 on the proliferation of preadipocytes assessed using an EdU assay and murine brown preadipocytes (WT-1 cells)[19,20] and human white preadipocytes (A41 hWAT-SVF cells)[21] (Fig. 1e). As the concentration of SerpinA1 added to brown preadipocytes increased the EdU[+]/DAPI[+] proliferating cell ratio increased by 1.7- to 2.2-fold (Fig. 1f, g). SerpinA1 also caused the proliferation of white preadipocytes in a dose-dependent manner (Fig. 1h, i). Thus, SerpinA1 exerts a stimulatory effect on the proliferation of both white and brown preadipocytes.

### SerpinA1 induces mitochondrial activity in both brown and white adipocytes

The effect of SerpinA1 on mature brown adipocytes function was further evaluated after induction of brown preadipocyte (WT-1 cell) differentiation (Fig. 2a). SerpinA1 did not affect either adipocyte differentiation, as assessed by Oil Red O staining (Supplementary Fig. 2a), or the mRNA expression of *Adiponectin, Leptin, FAS, ATGL, Glut4, PPARγ, AP2, CEBPα, Adrb3, Tfam, Elovl3, Cidea, PGC1α* and *PRDM16* (Fig. 2b). Interestingly, however, SerpinA1 upregulated the

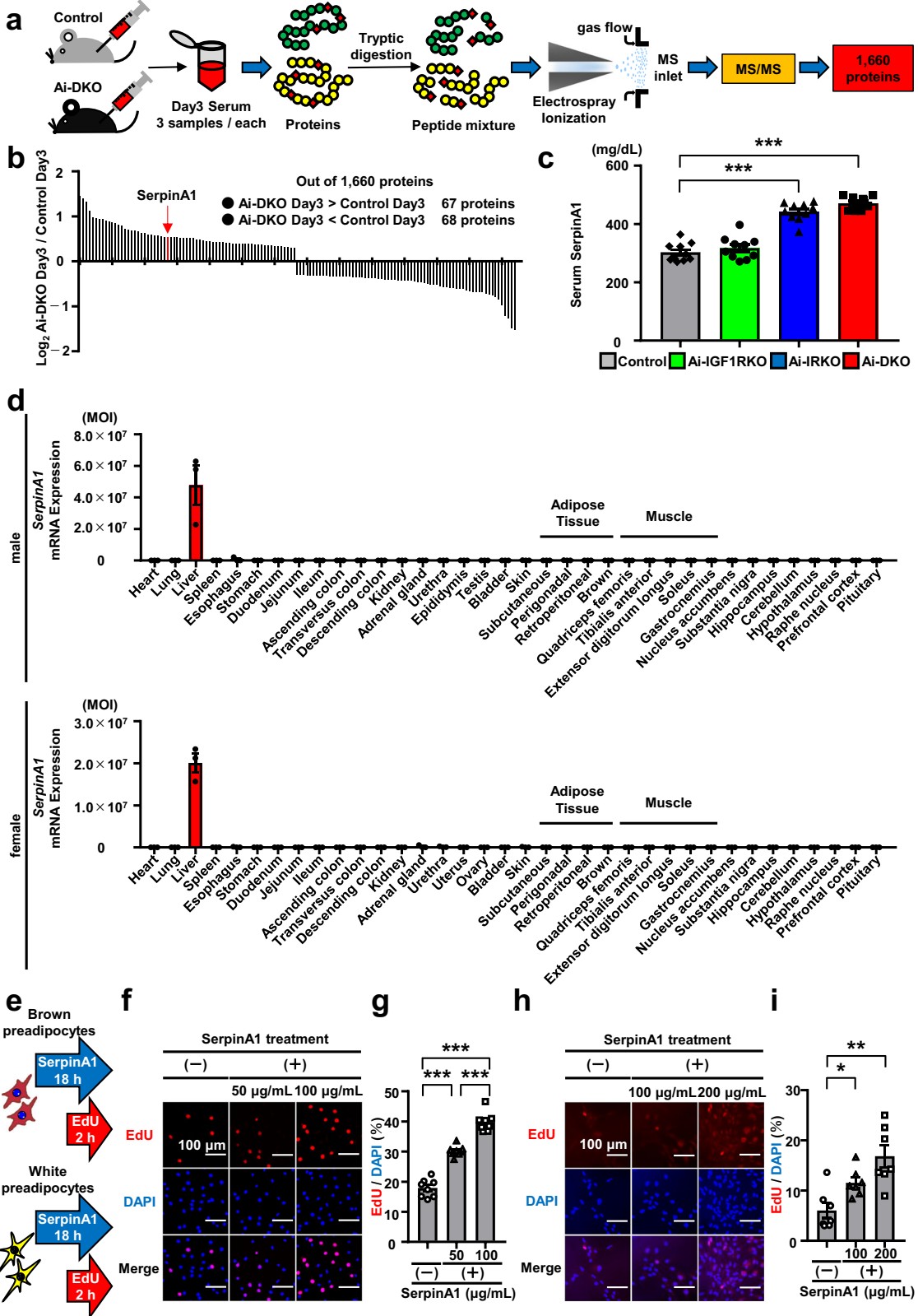

mRNA expression of the thermogenic marker *UCP1* by almost two-fold (Fig. 2b). This was accompanied by a parallel increase in the protein levels of UCP1 (Fig. 2c, d), as well as a significant increase in mitochondrial respiration, as measured by the maximal respiration oxygen consumption rate (OCR) and ATP production, without significant effects on other parameters, including basal respiration and spare respiratory capacity (Fig. 2e, f). Furthermore, UCP1 knockdown (UCP1-

KD) in mature brown adipocytes block the increase OCR by SerpinA1, indicating that the effect of SerpinA1 on adipocyte oxidative metabolism is primarily due to the upregulation of UCP1 (Supplementary Fig. 2b–d). SerpinA1 also stimulated the induction of UCP1 mRNA and protein in a dose-dependent manner, as well as increasing the expression of the beige adipocytes marker of mRNAs such as *DiO2, Pat2, CD137, CD40, CITED1* and *Sp100* in differentiating from white

**Fig. 1 | SerpinA1 induces preadipocyte proliferation. a** Workflow of the proteomics analysis of serum from control and Ai-DKO mice at 3 days after tamoxifen injection ($n = 3$) to identify factor(s) leading to preadipocyte proliferation. **b** Results of proteomics analysis performed according to the workflow in (**a**) and identification of SerpinA1. Statistical analysis was performed using two-tailed Student's $t$ test. $p < 0.05$, with an absolute |Log$_2$(fold change Ai-DKO/Control)| > 0.3. **c** Serum SerpinA1 levels in control, Ai-IGF1RKO, Ai-IRKO and Ai-DKO mice 3 days after tamoxifen injection. The data are presented as the mean ± SEM ($n = 10$, one-way ANOVA post hoc Bonferroni test, ***$p < 0.0001$). **d** *SerpinA1* mRNA expression in various organs and tissues of 4-month-old C57BL/6 mice with a primer that recognizes all five SerpinA1 paralogs. The data are presented as the mean ± SEM ($n = 3$).
**e** Schematic of the EdU assay for the analysis of preadipocyte proliferation after SerpinA1 treatment for 18 h. **f** Representative images of EdU-stained brown

preadipocytes (WT-1 cells) treated with different concentrations of SerpinA1 (0, 50, or 100 μg/mL). Untreated cells served as the negative Controls. The cells with red fluorescence are in S phase of mitosis, and all cells exhibit blue fluorescence. Scale bar = 100 μm. **g** Quantification of EdU-positive brown preadipocytes (WT-1 cells). The data are presented as the mean ± SEM ($n = 8$ technical replicates/group, one-way ANOVA post hoc Bonferroni test, ***$p < 0.0001$). **h:** Representative images of EdU-stained white preadipocytes (A41 hWAT-SVF cells) treated with different concentrations of SerpinA1 (0, 100, or 200 μg/mL). Scale bar = 100 μm.
**i** Quantification of EdU-positive white preadipocytes (A41 hWAT-SVF cells). The data are presented as the mean ± SEM ($n = 7$ technical replicates/group, two-tailed Student's $t$ test with Bonferroni's correction, *$p = 0.0101$ and **$p = 0.0015$). Source data are provided as a Source Data file.

preadipocytes (A41 hWAT-SVF cells) (Fig. 2g, h and Supplementary Fig. 2e–g). Likewise, SerpinA1 increased the mitochondrial respiration of mature white adipocytes, as measured by maximal respiration, ATP production and spare respiratory capacity, but not the basal respiration (Fig. 2i, j). The effects of SerpinA1 to increase UCP1 expression and mitochondrial activity in mature white adipocytes reflects the ability of this peptide to promote conversion of these cells to beige adipocytes.

## SerpinA1 forms a complex with EphB2 to promote preadipocyte proliferation

To determine how SerpinA1 regulates preadipocyte proliferation and mitochondrial activity, we investigated the proteins associated with SerpinA1 as potential components of the interaction complex in mature brown adipocytes. To this end, SerpinA1 was tagged with 3×Flag and expressed in brown preadipocytes (Fig. 3a–g). Mass spectrometry-based proteomics analysis following cell lysis and immunoprecipitation using anti-Flag beads (Fig. 3h and Supplementary Fig. 3a) revealed an average of 6,556 proteins in all samples, with 45 proteins differentially abundant between the SerpinA1-overexpressing (SerpinA1-OE) and control cells (Fig. 3i). EPH receptor B2 (EphB2) was identified as the top upregulated protein that interacts with SerpinA1 (Fig. 3i). Western blot analysis of the proteins immunoprecipitated with SerpinA1 confirmed that EphB2 was part of the complex of interacting proteins (Fig. 3j and Supplementary Fig. 3b).

Eph receptors belong to the superfamily of transmembrane tyrosine kinase receptors. Eph receptors are divided into the EphA or EphB subfamily depending on whether they preferentially bind to membrane-anchored or transmembrane ephrin ligands, referred to as ephrin-As and ephrin-Bs, respectively[22,23]. Eph receptors mainly affect the dynamics of cellular protrusions and cell migration by modifying cytoskeletal organization and cell adhesion and influence cell proliferation and cell fate determination[24]. EphB2 activation leads to the activation of FAK, potentially initiating downstream signaling pathways of EphB2[25]. Erk1/2 is also downstream of EphB2[26]. EphB2 promotes cutaneous squamous cell carcinoma (cSCC) cell proliferation via the Erk1/2 signaling pathway[27]. The Erk1/2 signaling pathway is also linked to brown preadipocyte proliferation[28]. However, there have been no reports on a direct relationship between EphB2 and brown preadipocyte proliferation.

To investigate whether the physical interaction of EphB2 with SerpinA1 is involved in the EphB2-mediated effect on brown preadipocytes, we generated EphB2 knockout (EphB2-KO) brown preadipocytes using CRISPR–Cas9 (Fig. 3k, l). We compared the proliferation rate of EphB2-KO brown preadipocytes with that of control preadipocytes in the presence and absence of SerpinA1. Control cells showed an -150% increase in proliferation rate after SerpinA1 treatment, while this SerpinA1-induced increase was abrogated in EphB2-KO cells (Fig. 3m, n). In EphB2-KO brown preadipocytes, the SerpinA1-induced elevation of FAK and Erk1/2 phosphorylation levels were also abrogated (Fig. 3o, p and Supplementary Fig. 3c, d). Thus,

SerpinA1 acts on EphB2 to enhance the phosphorylation of FAK and Erk1/2, and this acts as a proliferation signal in BAT.

## SerpinA1 induces adipocyte browning through interaction with EphB2

To investigate the role of EphB2 in mature brown adipocytes, we examined the expression of differentiation markers after brown adipocyte differentiation in EphB2-KO cells. Compared with control cells, EphB2-KO cells did not exhibit any differences in differentiation, as assessed by Oil Red O staining (Fig. 4a) or differences in the expression of *Adiponectin*, *Leptin*, *PPARγ*, or *Glut4*, but EphB2-KO cells showed a 66.8% decrease in the level of the thermogenic marker *UCP1* after adipogenic induction (Fig. 4b). EphB2 KO also abrogated the elevation of *UCP1* mRNA expression induced by SerpinA1 in differentiated adipocytes, demonstrating the role of EphB2 in the regulation of UCP1 expression through SerpinA1 (Fig. 4b). Likewise, when the expression of EphB2 was decreased using small interfering RNA (siRNA), there was also decreased browning of SerpinA1 treated adipocytes (Fig. 4c). Consistent with this, oxygen consumption rate (OCR), maximal respiration (23.8%) and spare respiratory capacity (26.8%) as assessed by Seahorse Analysis were significantly decreased, and both basal respiration and ATP production showed trends to a decrease in EphB2-KO cells compared with Controls (Fig. 4d, e). These results indicate that EphB2 plays important roles in adipocyte browning and preadipocyte proliferation induced by SerpinA1.

In brown adipocytes, β3-adrenergic receptor-mediated p38 mitogen-activated protein kinase (p38 MAPK) activation is also in involved in increased UCP1 expression[29], and p38 MAPK is a known downstream signal of EphB2[27,30]. In EphB2-KO mature brown adipocytes, the SerpinA1-induced elevation of FAK and p38 phosphorylation levels were abrogated (Fig. 4f, g and Supplementary Fig. 4a, b). Thus, SerpinA1 acts as a positive regulator of UCP1 expression and signaling in BAT by acting on EphB2 to enhance the phosphorylation of FAK and p38.

## Liver-specific SerpinA1-overexpressing transgenic mice exhibit increased browning and adaptive thermogenesis

To examine how SerpinA1 affects adipose tissues in vivo, we generated liver-specific SerpinA1-overexpressing transgenic (SPA1Tg) mice by transducing mice with the human *SerpinA1* gene with a 3×Flag tag downstream of the albumin promoter (Fig. 5a and Supplementary Fig. 5a–d). This produced only a modest elevation in SerpinA1, such that in SPA1Tg mice fed a chow diet (CD), the combined serum concentration of mouse and human SerpinA1 was elevated by approximately 1.3-fold above control animals, with mouse SerpinA1 levels around 300 mg/dl and human SerpinA1 levels around 400 mg/dl (Supplementary Fig. 5d). As a result, analysis of the livers of male SPA1Tg mice fed a CD showed no significant differences compared to Controls regarding weight, microscopic findings using HE-stained sections, or expression of mRNAs (Supplementary Fig. 5e–g). However, these both male and

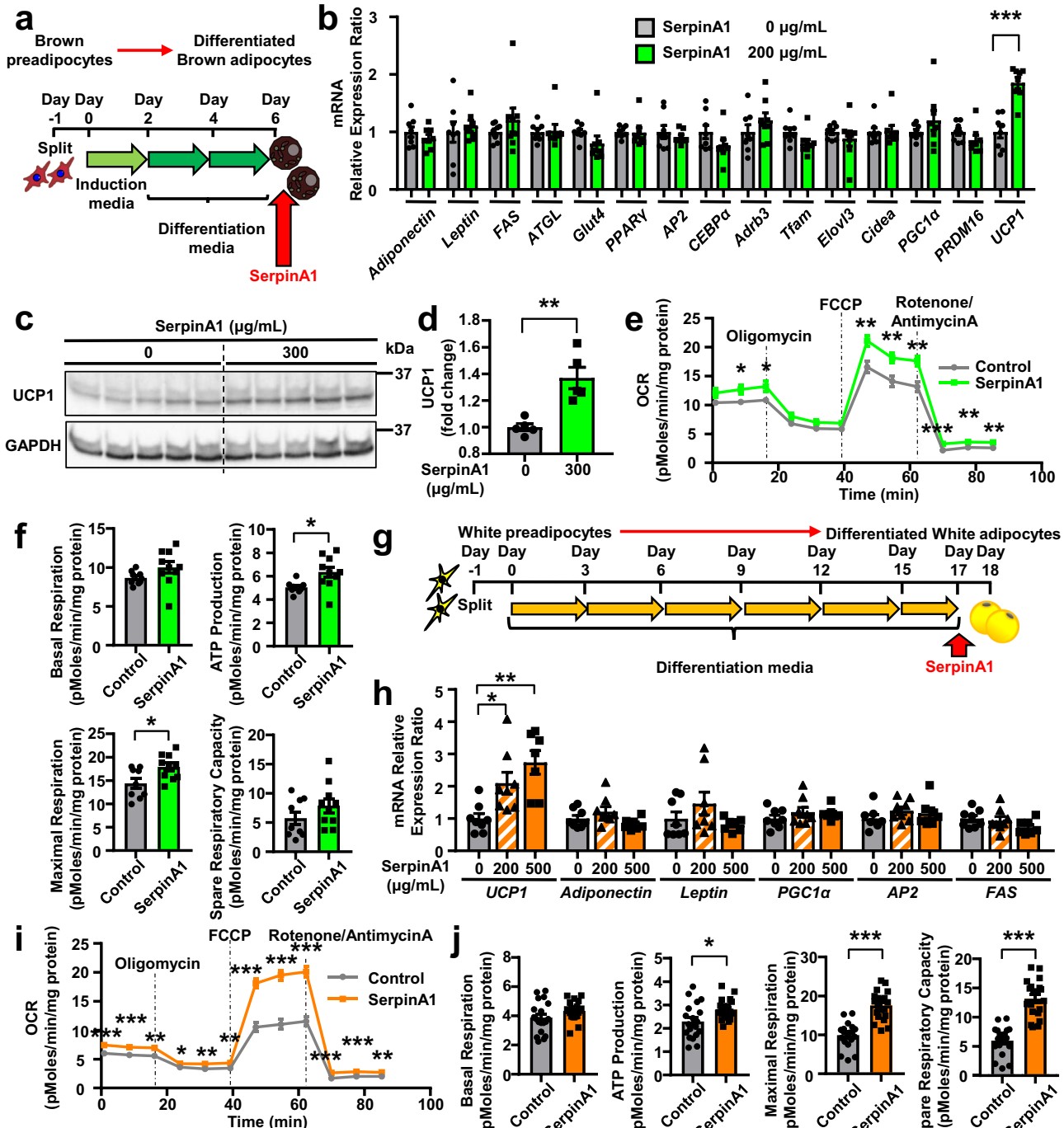

**Fig. 2 | SerpinA1 induces mitochondrial activity in both brown and white adipocytes. a** Schematic of the protocol used to induce brown preadipocytes to differentiate into mature brown adipocytes. **b** Relative mRNA expression of genes in mature brown adipocytes treated with 0 or 200 μg/mL SerpinA1 for 6 h ($n = 8$ technical replicates/group, ***$p < 0.0001$). **c** UCP1 protein expression in mature brown adipocytes treated with 0 or 300 μg/mL SerpinA1 for 24 h ($n = 5$). **d** Quantification of UCP1 protein levels in (**c**), expressed relative to the level of the protein standard ($n = 5$ technical replicates/group, **$p = 0.0025$). **e** Representative traces of the oxygen consumption rate (OCR) in mature brown adipocytes treated with 0 or 300 μg/mL SerpinA1 for 16 h (Control $n = 9$, SerpinA1 $n = 10$, technical replicates/group, *$p < 0.05$, **$p < 0.01$ and ***$p < 0.001$). **f** Quantitation of basal respiration, ATP production (*$p = 0.0141$), maximal respiration (*$p = 0.0185$) and spare respiratory capacity (Control $n = 9$, SerpinA1 $n = 10$, technical replicates/

group). **g** Schematic of the protocol used to induce white preadipocytes (A41 hWAT-SVF cells) to differentiate into mature white adipocytes. **h** Relative mRNA expression of genes in mature white adipocytes treated with different concentrations of SerpinA1 (0, 200, or 500 μg/mL) for 16 h (0 μg/mL $n = 8$, 200 μg/mL $n = 8$, and 500 μg/mL $n = 7$, technical replicates/group, *$p = 0.0444$ and **$p = 0.0017$). **i** Representative traces of the OCR in mature white adipocytes treated with 0 or 500 μg/mL SerpinA1 for 36 h (Control $n = 19$, SerpinA1 $n = 20$, technical replicates/group, *$p < 0.05$, **$p < 0.01$ and ***$p < 0.001$). **j** Quantitation of basal respiration, ATP production (*$p = 0.0132$), maximal respiration (***$p < 0.0001$) and spare respiratory capacity (***$p < 0.0001$) (Control $n = 19$, SerpinA1 $n = 20$, technical replicates/group). Data are presented as mean ± SEM. $P$ values were determined using two-tailed Student's $t$ test: (**b**, **d**–**f**, **i**–**j**); one-way ANOVA post hoc Bonferroni test: (**h**). Source data are provided as a Source Data file.

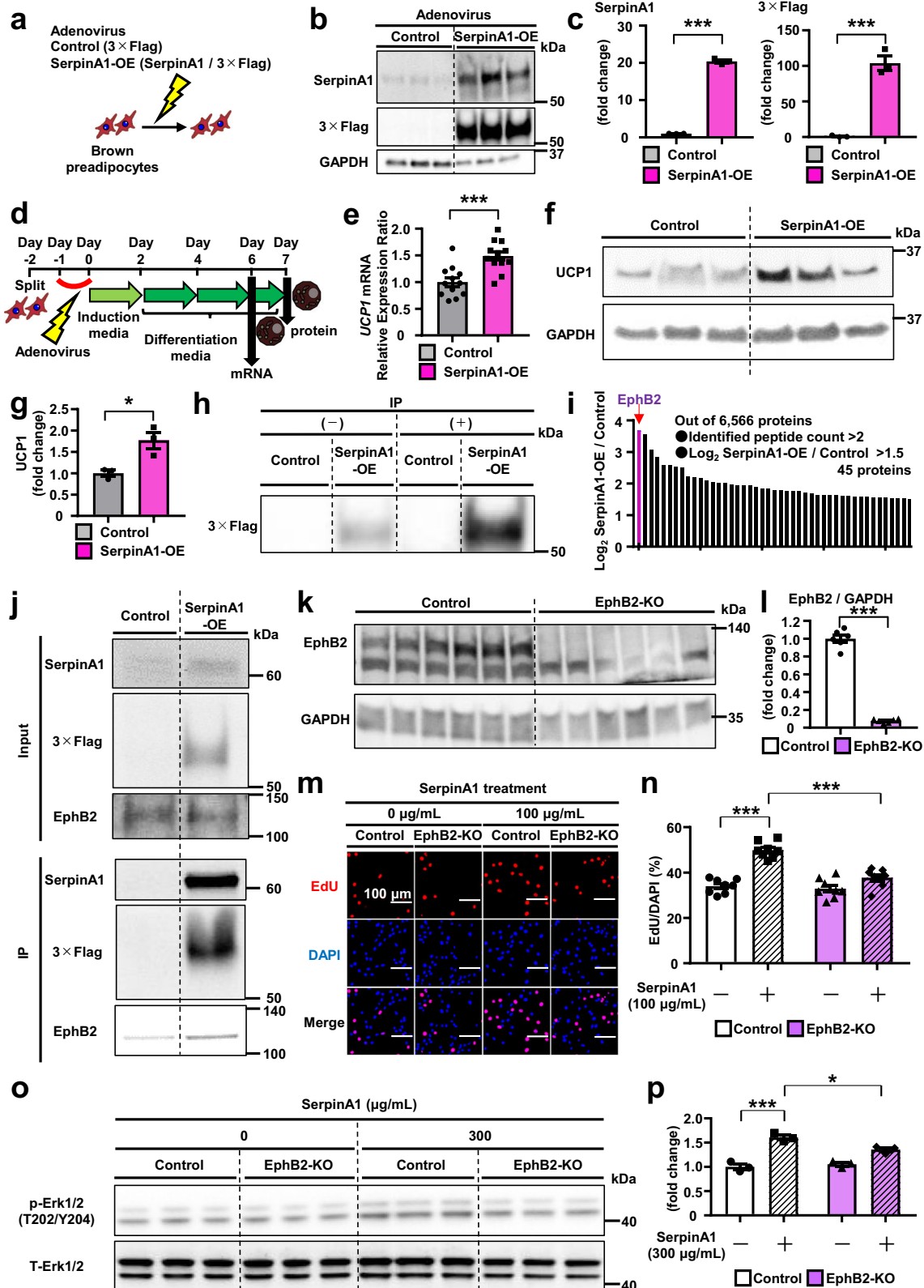

female SPA1Tg mice fed a CD exhibited reductions in inguinal WAT (19.1% in males and 33.9% in females adjusted for per body weight) (Fig. 5b and Supplementary Fig. 5h). Female mice, but not male mice, also exhibited 30.1% reduction in perigonadal fat, but neither had a reduction in BAT mass (Fig. 5b and Supplementary Fig. 5h). Consistent with these changes, male and female SPA1Tg mice exhibited significant decreases in adipocyte cell diameter in iWAT

and perigonadal WAT (pgWAT) and had smaller lipid droplets in BAT (Fig. 5c, d and Supplementary Fig. 5i), and whole-body micro-CT imaging showed a significant decrease in fat mass in both male and female 12-week-old SPA1Tg mice (Supplementary Fig. 5j). Lean body mass also trended to be slightly greater in both male and female SPA1Tg mice compared with Controls, but this difference was not significant (Supplementary Fig. 5j).

**Fig. 3 | SerpinA1 forms a complex with EphB2 to promote preadipocyte proliferation. a** Schematic of the protocol used to transduce brown preadipocytes with adenovirus carrying 3×Flag-tagged SerpinA1. **b** Adenovirus-mediated 3×Flag-tagged SerpinA1 protein overexpression brown preadipocytes ($n = 3$). **c** Quantification of protein levels in (**b**) ($n = 3$, ***$p < 0.0001$). **d** Schematic of the protocol used to induce the differentiation of brown preadipocytes infected with adenovirus carrying 3×Flag-tagged SerpinA1. **e** Relative *UCP1* mRNA expression in mature control and SerpinA1-OE brown adipocytes ($n = 12$, ***$p = 0.0004$). **f** UCP1 protein expression in mature control and SerpinA1-OE brown adipocytes ($n = 3$). **g** Quantification of protein levels in (**f**) ($n = 3$, *$p = 0.0200$). **h** Representative immunoblots before and after IP ($n = 1$) in control and SerpinA1-OE brown preadipocytes. **i** Potential SerpinA1-interacting proteins identified by mass spectrometry-based proteomics analysis followed by IP. **j** Identification of EphB2 as a member of the SerpinA1 complex via the coprecipitation of SerpinA1 and EphB2 by immunoblotting ($n = 1$). **k** EphB2 protein expression in control and CRISPR–Cas9-mediated EphB2-KO brown preadipocytes ($n = 6$). **l** Quantification of protein levels in (**k**) ($n = 6$, ***$p < 0.0001$). **m** Representative images of EdU-stained control and EphB2-KO brown preadipocytes treated with different concentrations of SerpinA1 (0 or 100 μg/mL, 18 h). Scale bar = 100 μm. **n** Quantification of EdU-positive control and EphB2-KO brown preadipocytes treated with different concentrations of SerpinA1 (0 or 100 μg/mL, 18 h) ($n = 8$, ***$p < 0.0001$). **o** The protein levels of p-ERK1/2 and T-ERK1/2 in control and EphB2-KO brown preadipocytes treated with different concentrations of SerpinA1 (0 or 300 μg/mL, 24 h) ($n = 3$). **p** Quantification of protein levels in (**o**) ($n = 3$, *$p = 0.0357$ and ***$p = 0.0001$). Data are presented as mean ± SEM. *P* values were determined using two-tailed Student's *t* test: (**c, e, g, l**); one-way ANOVA post hoc Bonferroni test: (**n, p**). All replicates performed in Fig. 3 were biologically independent cell clones/group. Source data are provided as a Source Data file.

Consistent with the increased activity of BAT, analysis of gene expression in BAT in male SPA1Tg mice showed a 2.5-fold increase in the expression of *UCP1* mRNA, with no change in the expression of *Adiponectin, Leptin, FAS, ATGL, Glut4, PPARγ, AP2, CEBPα, Adrb3, Tfam, Cidea, PGC1α* or *PRDM16* (Fig. 5e). *UCP1 mRNA* expression was also significantly increased by ~260% in iWAT in male SPA1Tg mice (Fig. 5f), and UCP1 protein, as measured by immunostaining, increased in both BAT and iWAT from both male and female SPA1Tg mice compared to Controls (Fig. 5g, h and Supplementary Fig. 5k), demonstrating the potent role of SerpinA1 in browning through the activation of UCP1 expression. The phosphorylation of p38 MAP kinase, a signal important for UCP1 expression, was increased in the BAT of male and female SPA1Tg mice compared to that of control mice (Fig. 5i, j and Supplementary Fig. 5l, m). Furthermore, unlike control mice whose core (rectal) and BAT (interscapular) temperatures fell upon cold exposure (4 °C) for 180 min, SPA1Tg mice maintained their core and BAT temperatures during cold exposure (Fig. 5k, l and Supplementary Fig. 5n). Thermal imaging also demonstrated a high body temperature over the interscapular region in both male and female SPA1Tg mice, consistent with increased activity (Fig. 5m and Supplementary Fig. 5o). Furthermore, 12-week-old CD-fed male SPA1Tg mice showed a significant increase in the OCR (VO₂), demonstrating increased energy expenditure (Supplementary Fig. 5p, q). These results show the biological function of SerpinA1 in inducing energy expenditure in adipose tissues in vivo.

To investigate the role of EphB2 in the function of SerpinA1 in vivo, we directly injected siEphB2 into the interscapular BAT of SPA1Tg mice. This led to an approximately 50% decrease in EphB2 protein and attenuated the elevated UCP1 protein expression in the BAT of male and female SPA1Tg (Supplementary Fig. 6a–c). To evaluate the effect of EphB2 knockdown in BAT on the in vivo function of SerpinA1, we examined heat production during 4 °C cold exposure for 4 days after injection. This revealed that the elevated rectal temperature in male and female SPA1Tg mice was attenuated by EphB2 knockdown in BAT, with results consistent with the body temperature over the interscapular region in thermal imaging (Supplementary Fig. 6d, e). Taken together, these results indicate that EphB2 plays an important role in heat production of BAT induced by SerpinA1 in vivo.

## SPA1Tg mice exhibit improved glucose metabolism

To define the impact of the increased BAT activity and beginning of WAT in SerpinA1 transgenic mice, we assessed the effects of the transgene on body weight and glucose metabolism. When fed a CD, control and SPA1Tg mice showed no differences in body weight, food intake, random blood glucose, or serum insulin levels (Supplementary Fig. 7a–d). However, both male and female SPA1Tg mice showed significantly lower fasting blood glucose levels (male 137.0 ± 1.7 mg/dl, female 121.3 ± 4.1 mg/dl) as compared to controls (male 152.8 ± 5.6 mg/dl, female 146.0 ± 4.3 mg/dl) (Fig. 6a and Supplementary Fig. 7e). Consistent with this, 12-week-old CD-fed male and female SPA1Tg mice

had better glucose tolerance and had significantly lower serum insulin levels than the Controls as measured by an intraperitoneal glucose tolerance test (ipGTT) (Fig. 6b, c and Supplementary Fig. 7f–i). The reduced insulin secretion can be attributed to improved insulin sensitivity, as assessed by intraperitoneal insulin tolerance testing (ipITT) in 12-week-old CD-fed male and female SPA1Tg mice (Fig. 6d, e and Supplementary Fig. 7j, k).

When the SPA1Tg mice were challenged with a 60% high-fat diet (HFD) for 3 months (Fig. 6f), the body weights of the SPA1Tg mice were similar to those of control mice fed the same diet up to 10 weeks of age, but by 17 weeks, the body weights of SPA1Tg mice were approximately 8% lower than that of the Controls (Fig. 6g). In HFD-fed male SPA1Tg mice, this was associated with decreased iWAT (23.1%) and pgWAT (28.9%) mass, but no significant change in BAT mass (Supplementary Fig. 7l). HFD-fed SPA1Tg mice also displayed lower fasting blood glucose levels (158.8 ± 8.8 mg/dl) than HFD-fed control mice (192.3 ± 10.9 mg/dl) (Fig. 6h). Likewise, compared to HFD-fed control mice, HFD-fed SPA1Tg mice showed improved glucose levels during an ipGTT and lower insulin resistance during an ipITT (Fig. 6i–l). Histologically, the adipocyte cell diameters in iWAT and pgWAT were significantly decreased in HFD-fed SPA1Tg mice compared with HFD-fed control mice, and multilocular lipid droplets were maintained in the BAT of HFD-fed SPA1Tg mice (Fig. 6m and Supplementary Fig. 7m). Importantly, UCP1 protein levels, as measured by immunostaining, in both iWAT and BAT were increased in HFD-fed SPA1Tg mice compared with HFD-fed control mice (Fig. 6n). Consistent with the increase in browning, these mice displayed cold resistance when placed in a 4 °C environment for 180 min (Fig. 6o) and maintained higher interscapular and rectal temperatures than in HFD-fed control mice (Fig. 6p, q).

## Quintuple SerpinA1a-e knockout (SPA1KO) mice exhibit decreased browning and impaired energy expenditure and glucose metabolism

To evaluate whether SerpinA1 is important for adipocyte function and metabolism in vivo, we generated whole-body SerpinA1 knockout mice. This is complicated since in wild-type (WT) mice, there are six *SerpinA1* paralogs (*SerpinA1a-f*) on chromosome 12 due to gene amplification events, and five of these *SerpinA1* genes (*SerpinA1a-e*) are expressed in the liver. Using CRISPR to target all five genes, we succeeded in generating a quintuple SPA1KO mouse model (Fig. 7a and Supplementary Fig. 8a). Indeed, SPA1KO mice showed an almost complete absence of *SerpinA1* mRNA in the liver and loss of SerpinA1 protein in both the liver and serum (Fig. 7b, c and Supplementary Fig. 8b). In humans, A1AT deficiency is a disease caused by mutations in the SerpinA1-coding sequence and is known to cause chronic obstructive pulmonary disease (COPD) and liver cirrhosis[31]. Interestingly, neither male nor female SPA1KO mice showed any macroscopic or microscopic changes in lung by HE-staining compared to Controls (Supplementary Fig. 8c, d). Likewise, livers of SPA1KO mice showed no significant differences compared to Controls regarding weight,

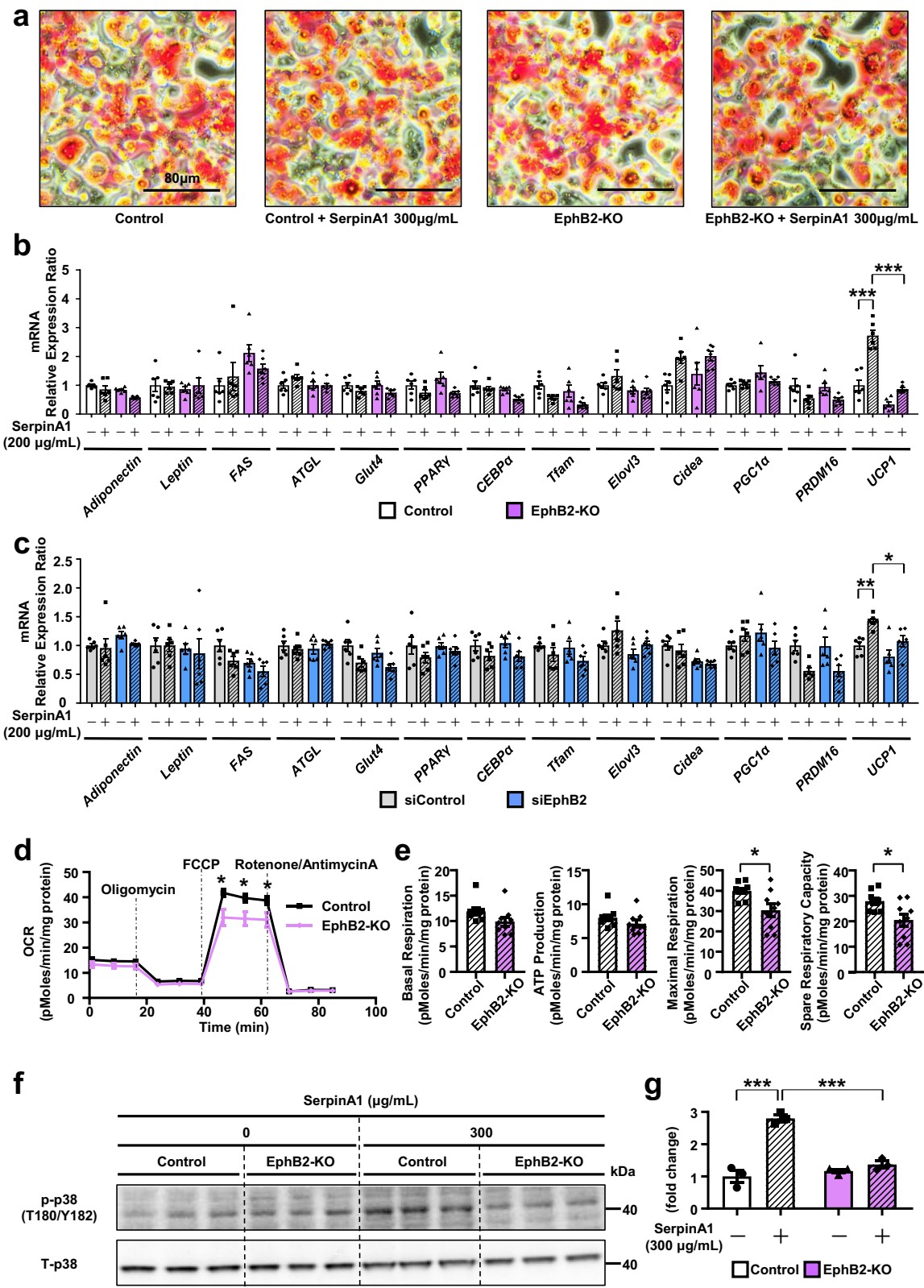

histologic appearance in HE-stained sections, or expression of mRNAs (Supplementary Fig. 8e, g). Furthermore, there were no changes in insulin signaling as measured by IR, Akt, or Erk1/2 phosphorylation in the liver (Supplementary Fig. 8h, i).

Despite the lack of changes in liver, male and female SPA1KO mice showed significantly increased weight of both iWAT and pgWAT, although there were no significant differences in body weight or food

intake (Supplementary Fig. 8j–l). Compared to control mice, SPA1KO mice displayed significantly greater adipocyte cell diameters in iWAT and pgWAT, as well as larger lipid droplets in BAT (Fig. 7d, e and Supplementary Fig. 8m). UCP1 expression, as measured by immunostaining, in both iWAT and BAT was decreased in male and female SPA1KO mice compared with Controls (Fig. 7f and Supplementary Fig. 9a). In addition, the levels of other thermogenic markers, such as

**Fig. 4 | SerpinA1 induces adipocyte browning through interaction with EphB2.**
**a** Oil Red O staining of control and EphB2-KO mature brown adipocytes treated with different concentrations of SerpinA1 (0 or 300 μg/mL) for 24 h. Scale bar = 80 μm. **b** Relative mRNA expression of genes in mature control brown adipocytes and brown adipocytes with CRISPR–Cas9-mediated EphB2 KO following treatment with different concentrations of SerpinA1 (0 or 200 μg/mL) for 12 h. The data are presented as the mean ± SEM ($n = 6$ biologically independent cell clones/group, one-way ANOVA post hoc Bonferroni test, ***$p < 0.0001$). **c** Relative mRNA expression of genes in mature control and EphB2-KD brown adipocytes electroporated with siRNA and treated with different concentrations of SerpinA1 (0 or 200 μg/mL) for 12 h. The data are presented as the mean ± SEM ($n = 6$ biologically independent cell clones/group, one-way ANOVA post hoc Bonferroni test, *$p = 0.0413$ and **$p = 0.0093$). **d** Representative traces of the OCR in mature control and EphB2-KO brown adipocytes treated with 0 or 300 μg/mL SerpinA1 for 16 h. The data are presented as the mean ± SEM ($n = 9$ biologically independent cell clones/group, two-tailed Student's $t$ test, *$p < 0.05$). **e** Quantitation of basal respiration, ATP production, maximal respiration (*$p = 0.0143$) and spare respiratory capacity (*$p = 0.0147$). The data are presented as the mean ± SEM ($n = 9$ biologically independent cell clones/group, two-tailed Student's $t$ test). **f** The protein level of p-p38 and T-p38 in control and EphB2-KO mature brown adipocytes treated with different concentrations of SerpinA1 (0 or 300 μg/mL, 24 h) ($n = 3$). **g** Quantification of p-p38 protein levels in (**f**), expressed relative to the T-p38 protein level. The data are presented as the mean ± SEM ($n = 3$ biologically independent cell clones/group, one-way ANOVA post hoc Bonferroni test, Control +SerpinA1(-) vs Control+SerpinA1(+): ***$p < 0.0001$, Control+SerpinA1(+) vs EphB2-KO+SerpinA1(+): ***$p = 0.0002$). Source data are provided as a Source Data file.

*PRDM16* and *Tfam*, in both iWAT and BAT were significantly decreased by 20–54% in SPA1KO mice compared with Controls (Fig. 7g and Supplementary Fig. 9b). SPA1KO mice also showed higher sensitivity to cold exposure (4 °C), with a marked decrease in body temperature to ~35 °C after 120 min (Fig. 7h and Supplementary Fig. 9c). Thermal imaging also showed a markedly lower temperature in the interscapular region of SPA1KO mice (Fig. 7i and Supplementary Fig. 9d).

To confirm the effect of SerpinA1 on adipose tissues regeneration, we crossed the SPA1KO mice with Ai-IRKO mice to generate Ai-IRKO mice without SerpinA1 (Ai-IRKO without SerpinA1). SerpinA1 knockout impaired the fat recovery potential in Ai-IRKO mice that developed tamoxifen-induced lipodystrophy 6 weeks after tamoxifen injection. Thus, Ai-IRKO without SerpinA1 mice had 42% decrease in BAT and and 38% decrease WAT as compared to Ai-IRKO mice 6 weeks after withdrawal of tamoxifen induction (Fig. 7j and Supplementary Fig. 9e). UCP1 protein expression in BAT of Ai-IRKO without SerpinA1 mice was reduced compared with that of Ai-IRKO mice (Supplementary Fig. 9f). In response to cold exposure at 4 °C on 6 weeks after cessation of tamoxifen injections, Ai-IRKO mice in the fat recovery process showed cold tolerance comparable to control mice, whereas Ai-IRKO without SerpinA1 mice exhibited significantly reduced cold tolerance compared to Ai-IRKO mice (Fig. 7k). Thermal imaging showed markedly lower temperature in the interscapular region of Ai-IRKO without SerpinA1 mice than Ai-IRKO mice with SerpinA1 pathway intact (Supplementary Fig. 9g). Thus, the loss of SerpinA1 reduces recovery of BAT and WAT after withdrawal of tamoxifen injections in mice with inducible adipose tissue IR knockout. Similarly, a lack of SerpinA1 impairs the recovery of BAT function.

SPA1KO mice showed a significant decrease in the OCR (VO₂), demonstrating impairment of energy expenditure due to lower UCP1 expression in adipocytes (Fig. 7l, m). At three months of age, SPA1KO mice also showed a metabolic syndrome, with mild hyperglycemia (fed glucose 10w 151.6 ± 4.7 mg/dl, 11w 150.4 ± 3.6 mg/dl) compared to Controls (fed glucose 10w 138.3 ± 4.3 mg/dl, 11w 140.7 ± 3.0 mg/dl) (Supplementary Fig. 9h), as well as severe glucose intolerance on ipGTT and marked insulin resistance on ipITT (Fig. 7n–q and Supplementary Fig. 9i–l).

When challenged with HFD beginning at 5 weeks of age of SPA1KO mice (Fig. 7r), the body weight of SPA1KO mice did not differ from that of control mice up to 11 weeks of age; however, by 12 weeks of age, SPA1KO mice developed significant weight gain despite showing no significant difference in food intake (Fig. 7s and Supplementary Fig. 10a). Accordingly, HFD-fed SPA1KO mice showed 1.2- to 1.4-fold increase in iWAT and pgWAT mass. HFD-fed SPA1KO mice also showed a 38% greater BAT mass than HFD-fed control mice (Fig. 7t, u), and macroscopically, the BAT of HFD-fed SPA1KO mice appeared whiter than that of Controls (Fig. 7t). Immunostaining of BAT and iWAT revealed enlarged lipid droplets and decreased UCP1 protein levels in HFD-fed SPA1KO mice compared to HFD-fed control mice (Fig. 7v and Supplementary Fig. 10b), as well as decreased expression levels of *PRDM16 and Tfam* (Supplementary Fig. 10c). In contrast, these mice showed greater *Leptin* expression than Controls, indicating whitening of brown adipocytes in the BAT of HFD-fed SPA1KO mice (Supplementary Fig. 10c). Consistent with the decreased expression of thermogenic markers in the BAT of HFD-fed SPA1KO mice, the body temperature of these mice (both rectally and interscapularly) was not maintained during cold exposure (Fig. 7w, x). SPA1KO mice fed a HFD also exhibited higher blood glucose levels during an ipGTT and reduced fall in blood glucose during an ipITT, indicative of impaired glucose tolerance and insulin resistance (Supplementary Fig. 10d–g). Thus, feeding SPA1KO mice a HFD accelerates the onset of obesity and diabetes due to impairment of energy expenditure in adipocytes.

## Discussion

Inducible adipocyte specific knockout of IR and both IR and IGF1R (Ai-DKO) provide unique models to study drivers of adipose development, since in both of these, once induction of knockout is stopped by withdrawal of tamoxifen, there is rapid regeneration of both white and brown fat. In this study, using proteomic analysis of serum from Ai-DKO mice, we have identified SerpinA1 as a key factor involved in the regeneration of WAT and BAT and the return of normal thermogenesis in these mice. SerpinA1 is a member of the Serpin family, a family of proteins that act as serine proteinase inhibitors[32,33]. In humans, serpins are classified as extracellular clade A molecules (thirteen members on chromosomes 1, 14 and X) and intracellular clade B serpins (thirteen members on chromosomes 6 and 18)[33,34]. The majority (27 out of 36) of serpins are inhibitory, but some possess non-inhibitory catalytic activity[33,34]. SerpinA1 has been shown to inhibit neutrophil elastase[33–35]. In humans, mutations in the SerpinA1-coding sequence present as a disease known as Alpha 1 anti-trypsin deficiency (A1ATD). A1ATD results in liver cirrhosis and chronic obstructive pulmonary disease (COPD)[31] through accumulation of misfolded, aggregated A1AT protein. The lack of A1AT in the systemic circulation, exacerbated by factors such as smoking, increases the susceptibility to lung injury, early-onset lung emphysema and chronic obstructive pulmonary disease[36]. Consequently, individuals with Alpha 1 anti-trypsin deficiency are susceptible to lung injury, but typically have normal liver function[36]. Likewise, A1AT KO mice spontaneously develop pulmonary emphysema with advancing age[37,38], But as in the present study, young adult SerpinA1 KO mice have no abnormalities in the liver or lungs. In humans, A1ATD has been treated with A1AT replacement therapy, and more recently with fazirsiran, a small interfering RNA (siRNA), that reduces the production of mutant α−1 antitrypsin protein[39].

Thus far, there are few reports regarding potential effects of SerpinA1 on metabolism, although SerpinA1 has been shown to inhibit obesity-induced neutrophil elastase and improve insulin sensitivity[35,40]. Another member of the Serpin family, SerpinA12, is also known as visceral adipose tissue-derived serine protease inhibitor (vaspin), and serves as an adipokine that participates in the development of insulin resistance, obesity, and inflammation by inhibiting kallikreins 7 and 14[41].

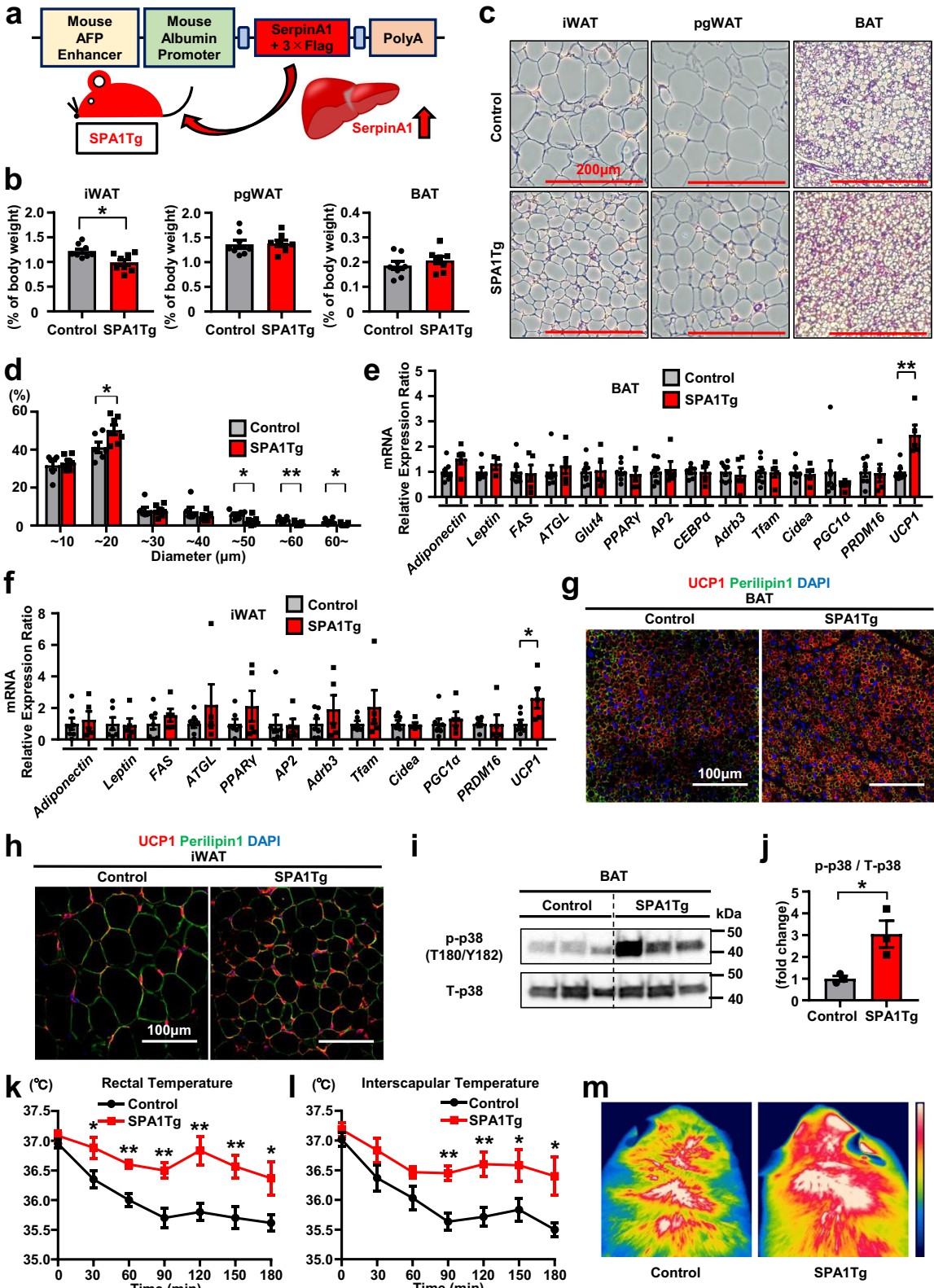

The pathophysiological mechanisms of metabolic syndrome are associated with dysregulation of multiple organ systems, including liver, adipose tissue, muscle and intestine. Whole-body metabolism is coordinated among tissues by many secreted factors, including peptide hormones, inflammatory mediators, signaling lipids, exosomal miRNAs, and other factors[3,14]. The liver is a source of multiple growth factors and hepatokines such as activin-

E, angiopoietin-like 4 (ANGPTL4), angiopoietin-like 6 (ANGPTL6), fibroblast growth factor 21 (FGF21) and growth differentiation factor 15 (GDF15), all of which have been observed to exert systemic effects and contribute to improvement of insulin resistance and enhancement of insulin sensitivity, although the molecular effects of these factors mostly remain to be identified[42]. We demonstrate that SerpinA1 is an important liver-derived secretory factor

**Fig. 5 | Liver-specific SerpinA1-overexpressing transgenic mice exhibit increased browning and adaptive thermogenesis.** All mice in Fig. 5 were male fed a chow diet. **a** Strategy for generating SPA1Tg mice. **b** Percentage of tissue weight per body weight of iWAT, pgWAT and BAT from 12-week-old control and SPA1Tg mice ($n = 8$, $*p = 0.0100$). **c** HE-stained sections of iWAT, pgWAT and BAT from 12-week-old control and SPA1Tg mice. Scale bars = 200 μm. **d** Diameter distribution of isolated iWAT from 12-week-old control and SPA1Tg mice (Control $n = 6$, SPA1Tg $n = 7$, $*p < 0.05$ and $**p < 0.01$). **e** Relative mRNA expression of genes in BAT from 12-week-old control and SPA1Tg mice (Control $n = 7$ and SPA1Tg $n = 5$; except for Control-*Leptin* ($n = 5$) and SPA1Tg-*Leptin* ($n = 3$), $**p = 0.0022$). **f** Relative mRNA expression of genes in iWAT from 12-week-old control and SPA1Tg mice (Control $n = 7$ and SPA1Tg $n = 5$; except for Control-*Leptin* ($n = 6$), Control-*Tfam* ($n = 6$) and Control-*PRDM16* ($n = 5$), $*p = 0.0220$). **g** Representative images of BAT sections

from 12-week-old control and SPA1Tg mice, immunostained for UCP1 and Perilipin1. **h** Representative images of iWAT sections from 12-week-old control and SPA1Tg mice, immunostained for UCP1 and Perilipin1. **i** The protein level of p-p38 and T-p38 in BAT from 8-week-old control and SPA1Tg mice ($n = 3$). **j** Quantification of protein levels in (**i**) ($n = 3$, $*p = 0.0310$). **k** Rectal temperatures of 18-week-old control and SPA1Tg mice exposed at 4 °C ($n = 6$, $*p < 0.05$ and $**p < 0.01$). **l** Interscapular temperatures of 18-week-old control and SPA1Tg mice exposed at 4 °C ($n = 6$, $*p < 0.05$ and $**p < 0.01$). **m** Thermal images showing the surface temperature over BAT in 18-week-old control and SPA1Tg mice at 180 min of exposure to 4 °C. The color bar indicates higher thermal temperatures towards the top. Data are presented as mean ± SEM. $P$ values were determined using two-tailed Student's $t$ test. Source data are provided as a Source Data file.

(hepatokine) that regulates adipocyte function and systemic metabolism.

Importantly, we have elucidated the molecular mechanism of SerpinA1 in the activation of brown adipocytes and shown that this occurs through binding to the EphB2 receptor and activation of the Ephrin pathway. EphB2 is a member of the Ephrin tyrosine kinase receptor family that bind Ephrin ligands. This is a large family of proteins including ten EphA and six EphB receptors, and six ephrin-A and three ephrin-B ligands[24,43,44]. EphA receptors preferentially bind glycosylphosphatidylinositol (GPI)-linked ephrin-A ligands, whereas EphB receptors bind transmembrane ephrin-B ligands[24,43]. The EphB2 receptor has the highest affinity for the ephrin-B2 ligand[45]. The Eph–ephrin complex depends on tyrosine phosphorylation and the associated proteins to generate bidirectional signals that affect both receptor-expressing and ephrin-expressing cells: Eph receptor forward signals and ephrin ligand reverse signals[46]. Eph receptors mainly affect the dynamics of cellular protrusions and cell migration by modifying cytoskeletal organization and cell adhesion, thus influencing cell proliferation and cell fate determination[24]. Among their many functions, signals mediated by EphA receptors and ephrin-A ligands regulate insulin secretion in pancreatic β-cells[47]. The ligand Ephrin-B1 (EFNB1) contributes to the suppression of the adipose inflammatory response[48]. In obesity, reductions in adipose EFNB1 levels accelerate the vicious cycle of adipose tissue inflammation[48]. The role of EphB2 in metabolism is less clear. EphB2 is cleaved at its extracellular domain near the N-terminus by a complex of coagulation factor VIIa (FVIIa) and tissue factor (TF), and this process reduces phosphorylated tyrosine signaling[49]. FVIIa is a circulating serine protease, and TF is its receptor. SerpinA1 possesses serine protease inhibitory activity and thus can potentially inhibit the protein cleavage through various kinds of serine proteases including FVIIa or other, as-of-yet, unidentified molecules. We show that SerpinA1 affects preadipocyte proliferation and mature adipocyte activation by increasing UCP1 expression and the maximal mitochondrial OCR through an EphB2-dependent mechanism. These results highlight an important role of liver - adipose tissue communication in maintenance of systemic metabolism.

Dietary induced obesity is associated with both enlargement of existing adipocytes and proliferation of white preadipocytes to form new adipocytes[50–52]. M2-like macrophages in adipose tissue have been shown to inhibit preadipocyte proliferation via the CD206/TGFβ signaling pathway[53]. In contrast, in BAT, β-adrenergic signaling and cold exposure can induce proliferation of brown adipocyte progenitor cells[51,54,55]. Several reports have explored the molecular mechanisms of induction of preadipocyte proliferation. Deletion of liver kinase b1 (Lkb1) in brown adipocytes may have paracrine or contact-dependent effects on preadipocyte proliferation[56]. Human brite/beige preadipocytes proliferate in response to pro-angiogenic factors in association with expanding capillary networks[57]. In BAT, stimulating the β3 adrenergic receptor activates the ERK MAPK pathway[58]. As in normal cells, sustained activation of Erk1/2 signaling regulates cell proliferation by affecting G1- to S-phase progression[59], and stimulation of Erk1/2

activation promotes proliferation of brown preadipocytes[28]. Erk1/2 and FAK are EphB2 downstream signals[26]. Although, there are no previous reports of EphB2 associated signals in brown preadipocytes, we demonstrate that the phosphorylation of FAK and Erk1/2 from the SerpinA1-EphB2 complex is enhanced in a pathway modulating brown preadipocyte proliferation.

In the classical thermogenesis pathway in brown adipose tissue, sympathetic neurons release noradrenaline, which activates β-adrenergic receptors, leading to activation of adenylyl cyclase (AC), an increase in the intracellular cyclic adenosine monophosphate (cAMP), activation of protein kinase A (PKA) phosphorylation and stimulation of lipolytic pathways[8,29,58]. PKA also activates the p38 MAPK pathway and upregulates the transcription of UCP1 and other thermogenic genes, including CCAAT-enhancer-binding proteins (C/EBPs), peroxisome proliferator-activated receptor γ (PPARγ), PPARγ coactivator-1α (PGC-1α), and PR domain zinc finger protein (PRDM16)[8,29,58]. Here, we demonstrate that SerpinA1 acting through the EphB2 can enhance the phosphorylation of p38 and serve as a upstream regulator of UCP1 in BAT.

SerpinA1Tg mice fed a CD also exhibit slight decreases in iWAT in both male and female mice and pgWAT in female mice. Although overall body weights did not differ from the controls on chow diet, whole-body micro-CT scan images showed a significant decrease in the fat mass of both male and female SPA1Tg mice. In addition, lean body mass was slightly higher in female and male SPA1Tg mice than in the control group, albeit not statistically significant. With high-fat diet feeding, however, there was a more significant decrease in body weight, which paralleled the lower weight of adipose tissues.

Importantly, SerpinA1 trangenic (SPA1Tg) mice show reduced fat mass and significantly increased levels of UCP1 in both WAT and BAT. SPA1Tg mice also exhibit improved glucose tolerance and increased heat production during cold exposure. Conversely, SerpinA1 knockout (SPA1KO) mice, in which all five SerpinA1 paralogs have been genetically inactivated, show adipocyte hypertrophy and significant lower levels of UCP1 expression. SPA1KO mice also exhibit impaired energy expenditure, leading to obesity, systemic insulin resistance and diabetes under HFD feeding. Thus, in vivo, SerpinA1 activates BAT and promotes browning of WAT to prevent obesity, increase energy expenditure and improve glucose tolerance, whereas the knockout of SerpinA1 does the converse. Taken together, our study demonstrates that liver-derived SerpinA1is an important circulating regulator of preadipocyte proliferation, mitochondrial activity and UCP1 expression in both BAT and WAT. As a result, higher levels of SerpinA1 promote energy expenditure and improve obesity and glucose metabolism in mice. Further study of the effects of SerpinA1 on adipocyte and glucose metabolism may open the shade of metabolic syndrome in humans.

## Methods

All experiments were performed in accordance with institutional ethical guidelines and approved by the licensing committee of

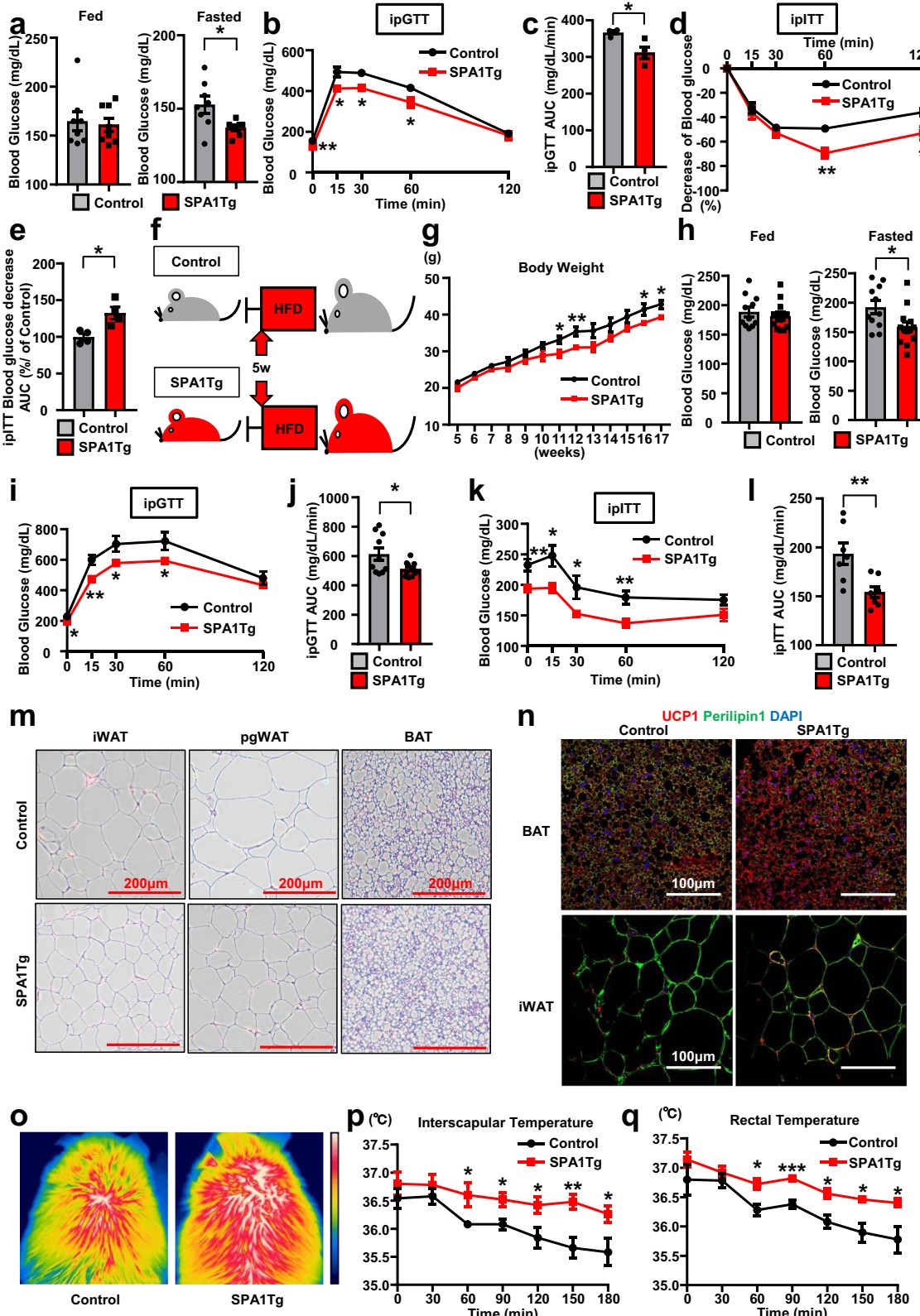

## Cell culture

The immortalized murine brown preadipocytes (WT-1) were derived from the stromal vascular fraction (SVF) of interscapular brown adipose tissue of newborn mice. The SVF cells were immortalized by SV40

T overexpression[19,20]. Immortalized human white preadipocytes (A41 hWAT-SVF) were derived from the SVF of superficial neck fat collected from a human male subject[21]. The SVF cells were immortalized by hTert overexpression. The immortalized murine brown preadipocytes (WT-1) have been deposited to Millipore Sigma and were authenticated by Millipore Sigma (#SCC255). The immortalized human white preadipocytes (A41 hWAT-SVF) have been deposited to ATCC and were

**Fig. 6 | SPA1Tg mice exhibit improved glucose metabolism. a–e** Results in 12-week-old male CD-fed control and SPA1Tg mice. **a** Fed and fasting blood glucose levels ($n = 8$, *$p = 0.0184$). **b** Results of the ipGTT ($n = 4$, *$p < 0.05$ and **$p < 0.01$). **c** Area under the curve (AUC) of the ipGTT in (b) ($n = 4$, *$p = 0.0168$). **d** Results of the ipITT ($n = 4$, *$p < 0.05$ and **$p < 0.01$). **e** AUC of the ipITT in (d) ($n = 4$, *$p = 0.0142$). **f** Schematic diagram of the HFD feeding program for male control and SPA1Tg mice beginning 5 weeks after birth. **g** Body weights of 5- to 17-week-old male HFD-fed control and SPA1Tg mice ($n = 7$, *$p < 0.05$ and **$p < 0.01$). **h–n** Results in 17-week-old male HFD-fed control and SPA1Tg mice. **h** Fed and fasting blood glucose levels (Control $n = 11$, SPA1Tg $n = 13$, *$p = 0.0244$). **i** Results of the ipGTT (Control $n = 10$, SPA1Tg $n = 11$, *$p < 0.05$ and **$p < 0.01$). **j** AUC of the ipGTT in (i) (Control $n = 10$, SPA1Tg $n = 11$, *$p = 0.0299$). **k** Results of the ipITT ($n = 7$, *$p < 0.05$ and **$p < 0.01$). **l** AUC of the ipITT in (k) ($n = 7$, **$p = 0.0088$). **m** HE-stained sections of iWAT, pgWAT and BAT. Scale bars = 200 μm. **n** Representative images of UCP1- and Perilipin1-immunostained BAT and iWAT sections. **o–q** Results in 20-week-old male HFD-fed control and SPA1Tg mice during 180 min exposed to 4 °C. **o**: Thermal images showing the surface temperature over BAT at 180 min. **p**: Interscapular temperatures ($n = 5$, *$p < 0.05$ and **$p < 0.01$). **q** Rectal temperatures ($n = 5$, *$p < 0.05$ and ***$p < 0.001$). Data are presented as mean ± SEM. $P$ values were determined using two-tailed Student's $t$ test. Source data are provided as a Source Data file.

authenticated by ATCC (#CRL-3386). Adeno-X™ 293 T cells (Takara, 632271) and Lenti-X™ 293 T cells (Takara, 632180) were purchased from Takara Bio Inc.

The cells were sorted and expanded in DMEM (Wako, 043-30085) supplemented with 10% heat-inactivated fetal bovine serum (FBS; Nichirei Biosciences Inc., 171012), 100 U/ml penicillin and 100 μg/ml streptomycin (Nacalai Tesque, Inc., 02892-54) at 37 °C in a 5% $CO_2$ incubator.

Brown preadipocytes were cultured with induction medium (DMEM containing 10% FBS (Nichirei Biosciences, Inc., 171012), 100 U/ml penicillin + 100 μg/ml streptomycin (Nacalai Tesque, Inc., 02892-54), 20 nM insulin (Sigma, I-9278), 1 nM triiodothyronine (T3, Sigma, T-6397), 1 μM dexamethasone (Sigma, D-4902), 500 μM iso-butylmethylxanthine (IBMX, Sigma, I-5879), 125 nM indomethacin (Sigma, I-7378) and 1 μM rosiglitazone (Sigma, R-2408)) for 2 days and then switched to another differentiation medium (DMEM containing 10% FBS, 1 nM T3 and 20 nM insulin), which was refreshed every 2 days, to induce differentiation into mature brown adipocytes[15,20,60].

White preadipocytes were cultured with differentiation medium (DMEM containing 10% FBS (Atlas, EF-0500-A), 100 U/ml penicillin + 100 μg/ml streptomycin (Nacalai Tesque, Inc., 02892-54), 0.5 μM insulin (Sigma, I-9278), 33 μM T3 (Sigma, T-6397), 0.1 μM dexamethasone (Sigma, D-4902), 500 μM IBMX (Sigma, I-5879), 30 μM indomethacin (Sigma, I-7378), 1 μM rosiglitazone (Sigma, R-2408), 33 μM biotin (Sigma, B-4639) and 17 μM pantothenate (Sigma, P-5155) for 18 days to induce differentiation into mature white adipocytes[21].

The concentrations of SerpinA1 used in our in vitro experiments (ranging from 50 to 500 μg/ml) were chosen based on its physiological blood levels observed in vivo, ensuring relevance to biological conditions.

### Adenoviral transduction of brown adipocytes

A fragment of SerpinA1 DNA tagged with 3×Flag amplified from a plasmid was inserted into an Adeno-X Adenoviral System 3 vector (CMV, Green) (Takara, 632267 Z2267N) according to the manufacturer's instructions. After transformation into *E. coli* Stellar™ competent cells (Takara, 636763) and screening for the constructs, the recombinant adenoviral vector was amplified, purified, and subsequently linearized with the restriction enzyme PacI (New England Biolabs, R0547L). Then, the vector was transduced into Adeno-X™ 293 T cells (Takara, 632271), and the presence of the adenovirus was confirmed with Adeno-X GoStix™ (Takara, 632270). The recombinant adenovirus overexpressing SerpinA1 was harvested and purified with an Add-N-Pure™ Adenovirus Purification Kit (Applied Biological Materials Inc., A910). The adenovirus was transduced into brown adipocytes using a CalPhos™ Mammalian Transfection Kit (Takara, 631312).

### Induction of EphB2 loss of function in brown adipocytes with siRNA

One hundred pmol (1 μM) of siRNA targeting mouse EphB2 (Dharmacon siGENOME Mouse Ephb2 (13844) siRNA - SMARTpool M-050820-01-0005) or control siRNA (Dharmacon siGENOME Non-Targeting siRNA Pool #1 D-001206-13-05) was electroporated into one million brown preadipocytes using a Nucleofector II device (Lonza Amaxa™ Nucleofector™ II) with a Cell Line Nucleofector™ Kit V (Lonza, VCA-1003). The next day, cell differentiation was induced as described above. On Day 6 of differentiation, the cells were treated with or without 200 μg/mL recombinant SerpinA1 protein (BioVision, 7294-1000) for 12 h and harvested for qRT–PCR. The sequences of the siRNAs were as follows:

* Dharmacon siGENOME SMARTpool siRNA D-050820-01, Ephb2: GAUCCAGUCUGUAGAGGUU
* Dharmacon siGENOME SMARTpool siRNA D-050820-02, Ephb2: ACUACGAGCUGCAGUACUA
* Dharmacon siGENOME SMARTpool siRNA D-050820-03, Ephb2: GGCAAGAUGUACUUCCAAA
* Dharmacon siGENOME SMARTpool siRNA D-050820-04, Ephb2: GCAGUACACCUUCGAGAUC
* Dharmacon siGENOME Non-Targeting siRNA Pool #1 D-001206-13-05: UAGCGACUAAACACAUCAA, UAAGGCUAUGAAGAGAUAC, AUGUAUUGGCCUGUAUUAG, and AUGAACGUGAAUUGCUCAA.

### CRISPR–Cas9-mediated EphB2 knockout in brown adipocytes

CRISPR–Cas9-mediated EphB2 knockout was achieved in a murine brown preadipocyte cell line (WT-1 cells) by lentiviral infection. LentiCRISPRv2 hygro (Addgene, 98291) was digested using BsmBI, and an annealed pair of oligos (forward: 5′-CACCGCGGCCAGATTGTCAA-CACGC-3′, reverse: 5′-AAACGCGTGTTGACAATCTGGCCGC-3′) was cloned into the single guide RNA scaffold. This plasmid was cotransduced into Lenti-X™ 293 T cells (Takara, 632180 Z2180N) with the viral packaging vectors psPax2 (Addgene, 12260) and pMD2.G (Addgene, 12259) using FuGENE® HD Transfection Reagent (Promega, E2311) according to the manufacturer's instructions. After transducing the resulting lentivirus into brown preadipocytes, hygromycin was used to select EphB2-KO cells. The protocol was performed as described in previous publications[61] and the Zhang Lab GeCKO website (http://www.genome-engineering.org/gecko/).

### CRISPR–Cas9-mediated UCP1 knockdown in brown adipocytes

CRISPR–Cas9-mediated UCP1 knockdown was achieved in a murine brown preadipocyte cell line (WT-1 cells) by lentiviral infection. LentiCRISPRv2 hygro (Addgene, 98291) was digested using BsmBI, and an annealed pair of oligos (forward: 5′-CACCGATCGCACAGCTTG GTACGCT-3′, reverse: 5′-AAACAGCGTACCAAGCTGTGCGATC-3′) was cloned into the single guide RNA scaffold. From this point on, the method was the same as that described above for "CRISPR-Cas9-mediated EphB2 knockout in brown adipocytes".

### Immunoprecipitation

Brown preadipocytes were transduced with control adenovirus or adenovirus overexpressing 3×Flag-tagged SerpinA1 for 24 h. After being cultured in normal medium for 2 days, these cells were rinsed once with PBS and treated with the chemical crosslinker 1 mM 3,3′-dithiobis (sulfosuccinimidyl propionate) (DTSSP) (Thermo Scientific™, 21578) in PBS on ice at 4 °C for 1 h. They were washed with stop solution containing 20 mM Tris pH 7.5 (Sigma, T2319) and 100 mM NaCl (Nacalai Tesque, Inc., 06900-14) on ice 3 times for 15 min each. The

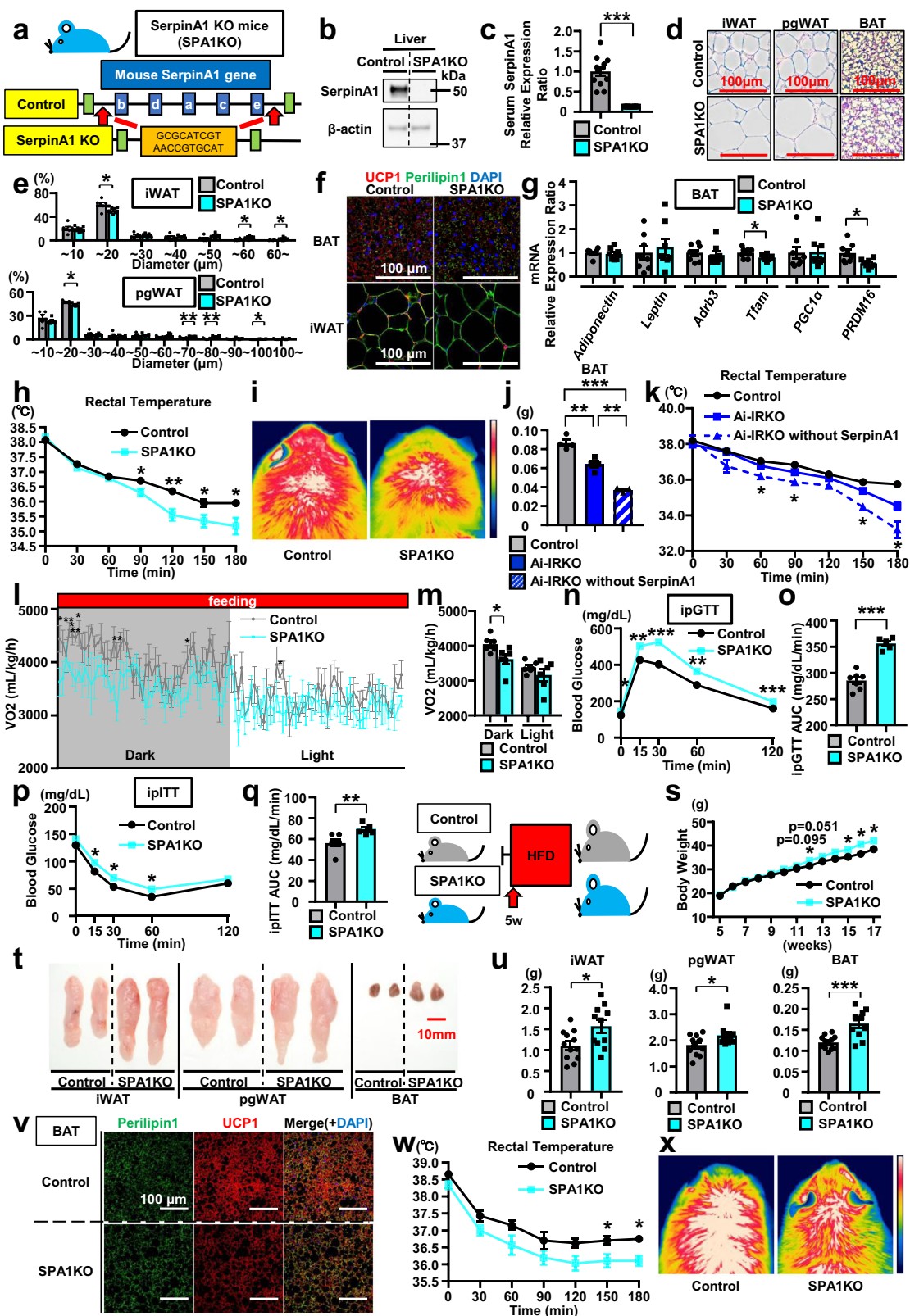

cells were harvested and lysed with lysis buffer consisting of 50 mM Tris (pH 7.5) (Sigma, T2319), 1 mM EDTA (Nacalai Tesque, Inc., 06894-85), 150 mM NaCl (Nacalai Tesque, Inc., 06900-14), 1% Triton X-100 (Sigma, T8787), protease inhibitor cocktail (Bimake, B14002-BIT) and phosphatase inhibitor cocktail A and B (Bimake, B15002-BIT). For 3×Flag-specific immunoprecipitation, anti-Flag-M2-conjugated beads (Sigma, M8823) were added to the cell lysate, and the sample was

slowly rotated at 4 °C for 1.5 h. After 3 washes with wash buffer consisting of 50 mM Tris (pH 7.5) (Sigma, T2319), 1 mM EDTA (Nacalai Tesque, Inc., 06894-85), 300 mM NaCl (Nacalai Tesque, Inc., 06900-14), 1% Triton X-100 (Sigma, T8787), protease inhibitor cocktail (Bimake, B14002-BIT) and phosphatase inhibitor cocktails A and B (Bimake, B15002-BIT), the samples were separated and eluted with 150 ng/μL 3×Flag peptide (Sigma, F4799) at 4 °C for 30 min. The

**Fig. 7 | Quintuple SerpinA1a-e knockout (SPA1KO) mice exhibit decreased browning and impaired energy expenditure and glucose metabolism.**
**a** Generation of SPA1KO mice. **b–g** 12-week-old male CD-fed control and SPA1KO.
**b** SerpinA1 protein ($n = 1$). **c** ELISA of serum SerpinA1 ($n = 12$, ***$p < 0.0001$). **d** HE sections. Scale bars = 100 μm. **e** iWAT (Control $n = 6$, SPA1KO $n = 9$) and pgWAT diameter ($n = 5$) (*$p < 0.05$ and **$p < 0.01$). **f** UCP1-immunostained BAT and iWAT sections. Scale bars = 100 μm. **g** mRNA expression in BAT ($n = 8$, *$p < 0.05$). **h–i** Cold exposure of 11-week-old male CD-fed control and SPA1KO at 4 °C during 180 min.
**h** Rectal temperatures ($n = 11$, *$p < 0.05$ and **$p < 0.01$). **i** Thermal images over BAT at 120 min. **j** Weight of BAT from male control, Ai-IRKO and Ai-IRKO without SerpinA1 (Control and Ai-IRKO $n = 4$, Ai-IRKO without SerpinA1 $n = 3$, **$p < 0.01$ and ***$p < 0.001$). **k** Rectal temperatures of male control, Ai-IRKO and Ai-IRKO without SerpinA1 exposed to 4 °C (Control and Ai-IRKO $n = 4$, Ai-IRKO without SerpinA1 $n = 3$, Ai-IRKO vs Ai-IRKO without SerpinA1, *$p < 0.05$). **l, m** 11-week-old male CD-fed control and SPA1KO. **l** VO$_2$ ($n = 6$). **m** Mean VO$_2$ in (**l**) ($n = 6$, *$p = 0.0268$). **n–q** 12-week-old male CD-fed control and SPA1KO (Control $n = 7$, SPA1KO $n = 5$). **n** ipGTT (*$p < 0.05$, **$p < 0.01$ and ***$p < 0.001$). **o** AUC of the ipGTT in (**n**) (***$p < 0.0001$). **p** ipITT (*$p < 0.05$). **q** AUC of the ipITT in (**p**) (**$p = 0.0096$). **r** HFD feeding program for male control and SPA1KO mice beginning 5 weeks old. **s** Body weights of 5- to 17-week-old male control and SPA1KO mice fed a HFD for 12 weeks (Control $n = 11$, SPA1KO $n = 9$, *$p < 0.05$). **t–x** 17-week-old male HFD-fed control and SPA1KO.
**t** Representative picture of adipose tissues (iWAT, pgWAT and BAT). Scale bar = 10 mm. **u** Weights of iWAT (*$p = 0.0158$), pgWAT (*$p = 0.0314$) and BAT (***$p = 0.0003$) (Control $n = 12$, SPA1KO $n = 11$). **v** UCP1-immunostained BAT sections. Scale bars = 100 μm. **w, x** 17-week-old male HFD-fed control and SPA1KO during 180 min exposed to 4 °C. **w** Rectal temperatures (Control $n = 4$, SPA1KO $n = 3$, *$p < 0.05$). **x** Thermal images at 180 min. Data are presented as mean ± SEM. $P$ values were determined using two-tailed Student's $t$ test: (**c, e, g, h, l–q, s, u, w**); one-way ANOVA post hoc Bonferroni test: (**j, k**). Source data are provided as a Source Data file.

immunoprecipitate was centrifuged on an Amicon Ultra4 (10 kDa) column (Millipore, UFC801008) at 4 °C for approximately 30 min, concentrated 7- to 8-fold, and then subjected to proteomics analysis. The control and SerpinA1-OE samples were used for liquid chromatography–tandem mass spectrometry (LC–MS/MS)-based proteomics analysis, for silver staining with a Pierce™ Silver Stain Kit (ThermoScientific™, 24612) after immunoblotting and western blotting.

Mouse liver and BAT complex sample lysates were prepared using cell lysis buffer (Cell Signaling, #9803) supplemented with 1 mM PMSF, and the concentration was adjusted to 1 mg/mL. Anti-SerpinA1 antibody (Proteintech Group, Inc., 16382-1-AP, 4.0 μg for 3.0 mg of total protein lysate) was added to 1 mL of the precleared lysate and incubated overnight at 4 °C with rotation to form the immunocomplex. The lysate and antibody (immunocomplex) mixtures were added to washed Dynabeads Protein A (Dynabeads, DB10001) and Dynabeads Protein G (Dynabeads, DB10003) pellets, which were subsequently incubated at room temperature for 1.5 h with rotation. The bead pellets were washed five times with 500 μL of wash buffer, resuspended in 30 μL of 4x SDS sample buffer and heated to 95 °C for 5 min. After the beads were removed, the samples were loaded onto an SDS-PAGE gel and analyzed by western blotting.

### LC–MS/MS-based proteomics
80 μL of serum samples from the WT Day 3 and Ai-DKO Day 3 conditions were subjected proteomics using LC-MS/MS as detailed in the previous paper[16].

After extracting protein from equal amounts of control and SerpinA1-OE immunoprecipitates, data-independent acquisition (DIA)-based proteomics analysis was outsourced to Kazusa Genome Technologies Inc. The protocol was performed as described in previous publications[20,62,63].

The eluted sample was added acetone (final acetone concentration 80% v/v) and incubated for 2 h at −20 °C. After removing the supernatant by centrifugation at $15,000 \times g$ for 15 min at 4 °C, the precipitate was redissolved in 0.5% sodium dodecanoate and 100 mM Tris-HCl, pH 8.5 by using a water bath-type sonicator (Bioruptor UCD-200 SonicBio Corp., Kanagawa, Japan). The pretreatment of shotgun proteome analysis was performed[20]. Peptides were directly injected onto a 75 μm × 20 cm, PicoFrit emitter packed in house with 2.7 μm core shell C18 particles at 45 °C and then separated with an 80 min gradient at a flow rate of 100 nl/min using an UltiMate 3000 RSLCnano LC system (Thermo Fisher Scientific, Waltham, MA, USA). Peptides eluting from the column were analyzed on a Q Exactive HF-X (Thermo Fisher Scientific) for overlapping window DIA-MS. MS1 spectra were collected in the range of 495–785 m/z at 30,000 resolution to set an automatic gain control target of 3e6 and maximum injection time of 55. MS2 spectra were collected in the range of more than 200 m/z at 30,000 resolution to set an automatic gain control target of 3e6,

maximum injection time of "auto.", and stepped normalized collision energy of 22, 26, and 30%. An isolation width for MS2 was set to 4 m/z and overlapping window patterns in 500–780 m/z were used window placements optimized by Skyline v4.1.

MS files were searched against a mouse spectral library using Scaffold DIA (Proteome Software, Inc., Portland, OR). The spectral library was generated from mouse protein sequence database (UniProt id UP000000589, reviewed, canonical) by Prosit[62]. The scaffold DIA search parameters were as follows: experimental data search enzyme, trypsin; maximum missed cleavage sites, 1; precursor mass tolerance, 8 ppm; fragment mass tolerance, 10 ppm; static modification, cysteine carbamidomethylation. The protein identification threshold was set both peptide and protein false discovery rates of less than 1%. Peptide quantification was calculated by EncyclopeDIA algorithm[63] in Scaffold DIA. For each peptide, the four highest-quality fragment ions were selected for quantitation. Protein quantification was estimated from the summed peptide quantification data.

### EdU cell proliferation assay
Brown and white preadipocytes were incubated with recombinant SerpinA1 protein (BioVision, 7294-1000) for 18 h in the absence of FBS and stained with EdU for the last 2 h. Recombinant SerpinA1 protein (BioVision, 7294-1000) was added to brown preadipocytes at a concentration of 0, 50, or 100 μg/mL; to brown preadipocytes with EphB2 knockout at concentrations of 0 or 100 μg/mL; and to white preadipocytes at a concentration of 0, 100, or 200 μg/mL. The cells were stained with a Click-iT™ Plus EdU Alexa Fluor™ 594 Imaging Kit (Invitrogen, C10639) and imaged with a confocal laser scanning microscope (OLYMPUS, FV3000). An All-in-One Fluorescence Microscope (Keyence, BZ-9000) and a BZ-II Dynamic Cell Count Analyzer (Keyence, BZ-H1CE) were used to quantify the ratio of EdU to DAPI staining.

### qRT–PCR
Total RNA was extracted from tissues with an RNeasy Mini Kit (Qiagen, California, USA), and complementary DNA (cDNA) was synthesized with ReverTra Ace® qPCR RT Master Mix (Toyobo, Osaka, Japan). Real-time PCR was performed using GoTaq qPCR Master Mix (Promega, Mannheim, Germany) on a QuantStudio™ 12 K Flex Real-Time PCR System (Applied Biosystems, CA, USA). All expression data were normalized to TBP or 18S expression. The amplification of specific transcripts was confirmed by analyzing melting curve profiles at the end of each PCR run. The sequences of all primers used for qRT–PCR in this study are shown in the Supplementary Table 1, 2.

### Immunoblotting
Cells were lysed in 1x RIPA lysis buffer (Merck Millipore, 20-188) supplemented with 0.1% sodium dodecyl sulfate (SDS), protease inhibitor cocktail (Bimake, B14002-BIT) and phosphatase inhibitor cocktails A and B (Bimake, B15002-BIT). Tissue samples were homogenized with

metal beads using a bead crusher (TAITEC, μT-12) in the same buffer. After the protein concentration was equalized, 4x NuPAGE™ LDS Sample Buffer (Invitrogen, NP0007) containing 5% 2-mercaptoethanol was added, and the mixture was boiled at 95 °C for 5 min. The proteins were separated by SDS–polyacrylamide gel electrophoresis and transferred onto polyvinylidene fluoride membranes.

The membranes were blocked in StartingBlock™ T20 (PBS) blocking buffer (Thermo Fisher, 37539) at room temperature for 1 h and probed with the appropriate primary antibodies in StartingBlock™ T20 (PBS) blocking buffer overnight at 4 °C. The primary antibodies for phospho-IR/IGF1R, 19H7 (#3024, 1:1000), IRβ, 4B8 (#3025, 1:1000), phospho-p44/42 MAPK (Erk1/2) (Thr202/Tyr204) (#9101, 1:1000), p44/42 MAPK (Erk1/2) (#9102, 1:1000), phospho-Akt (S473) (#9271, 1:1000), Akt, 11E7 (#4685, 1:1000), phospho-p38 MAPK, D3F9 (Thr180/Tyr182) (#4511, 1:1000), p38 MAPK (#9212, 1:1000), phospho-FAK (Tyr397) (#3283, 1:1000), FAK (#3285, 1:1000), GAPDH, D4C6R (#97166, 1:1000), β-Actin, 13E5 (#4970, 1:1000) and α-Tubulin (#2144, 1:1000) were from Cell Signaling Technology. SerpinA1 (alpha 1 Antitrypsin) (ab166610, 1:1000), SerpinA1 (alpha 1 Antitrypsin) (ab205152, 1:1000) and UCP1 (ab10983, 1:1000) antibody were from Abcam. EphB2 (AF467, 1:1000) antibody was from R&D Systems, Inc. Flag (F3040, 1:1000) antibody was from Sigma. The membranes were washed three times with Tris-buffered saline with Tween® 20 (TBS-T) (Takara, T9142) and incubated with an HRP-conjugated secondary antibody (1:1000) in StartingBlock™ T20 (PBS) blocking buffer for 1 h. Anti-rabbit IgG, HRP-linked antibody (Cell Signaling Technology, #7074, 1:1000), anti-mouse IgG, HRP-linked antibody (Cell Signaling Technology, #7076, 1:1000), anti-rat IgG, HRP-linked antibody (Cell Signaling Technology, #7076, 1:1000) and mouse anti-goat IgG-HRP (Santa Cruz, sc-2354, 1:1000) were used as secondary antibodies. The antibody signals were detected with Immobilon Western Chemiluminescent HRP Substrate (Millipore, Massachusetts, USA, WBKLS0500) according to the manufacturer's instructions. Cooled CCD Camera System Light-Capture II (ATTO), WSE-6170 LuminoGraph I CMOS (ATTO) and ChemiDoc™ Touch Gel Imaging System (BIO-RAD) was used to acquire western blot images. We quantified these with ImageJ (1.52a). All uncropped and unprocessed scans of the blots are provided in the Source Data file or as a Supplementary Fig. in the Supplementary Information.

### Measurement of the oxygen consumption rate of adipocytes with a Seahorse Bioanalyzer

The oxygen consumption rate (OCR) was assessed using a Seahorse XFe24 Flux Analyzer (Agilent Technologies) and a Mito Stress Test Kit according to the manufacturer's protocol.

Brown preadipocytes were plated in XFe24 cell culture microplates at a density of 8000 cells/well and induced to differentiate as described above. On Day 6, the differentiated brown adipocytes were treated with or without 300 μg/mL recombinant SerpinA1 protein (BioVision, 7294–1000) for 16 h. The cell medium was replaced with Seahorse base medium without phenol red, after which 2 μM oligomycin, 1.2 μM FCCP, and 1 μM rotenone/antimycin A were added in that order. Brown adipocytes in which EphB2 was knocked out were treated in the same manner except that the preadipocytes were seeded at a density of 2000 cells/well. Brown adipocytes in which UCP1 was knocked down were also treated in the same manner except that the preadipocytes were seeded at a density of 5000 cells/well.

White preadipocytes were plated in XFe24 cell culture microplates at a density of 10,000 cells/well and induced to differentiate as described above. On Day 16, the differentiated white adipocytes were treated with or without 500 μg/mL recombinant SerpinA1 protein (BioVision, 7294–1000) for 36 h. The cell medium was replaced with Seahorse base medium without phenol red, after which 2 μM oligomycin, 1.5 μM FCCP, and 1 μM rotenone/antimycin A were added in that order.

Basal respiration, ATP production, and maximal respiration capacity were calculated by subtracting nonmitochondrial respiration. The data for all wells at the three time points were averaged and used to create bar graphs. The cells were lysed in 0.1% SDS, and their protein concentrations were measured and used for the normalization of OCR values.

### Mice

Mice were housed at 20–22 °C on a 12 h light/dark cycle with average 50% Humidity and fed either a CD (#Rodent Diet CE-2, CLEA, Tokyo, Japan) or HFD (60% calories from fat; #HFD32, CLEA, Tokyo, Japan) in the animal facility at Kumamoto University, Japan. All experiments with research animals were performed in accordance with institutional ethical guidelines and approved by the licensing committee of Kumamoto University (Approval Numbers: A30-051, 2020-099, 2022-077 and A2024-092). Both sexes were considered in the study design and analysis.

4-month-old male and female C57BL/6 mice were analyzed in this study (3 per/each). Both liver-specific SerpinA1-overexpressing transgenic (SPA1Tg) mice (8 to 20-weekold male and 12-weekold female) and whole-body SerpinA1 knockout (SPA1KO) mice (11 to 17-weekold male and 12-weekold female) were generated and analyzed by us at Kumamoto University. Adiponectin-CreERT2 mice (10-weekold male, stock no. 025124) were purchased from Jackson Laboratories. Control (2-monthold male), Ai-IGF1RKO (2-monthold male), Ai-IRKO (2-monthold male) and Ai-DKO (2-monthold male) mice were maintained on a mixed (C57Bl/6 − 129 Sv) background by breeding Adiponectin-CreERT2 IRf/f and/or IGF1Rf/f with IRf/f and/or IGF1Rf/f mice[15,60]. Control (2 to 3-monthold male), Ai-IRKO (2 to 3-monthold male) and Ai-IRKO without SerpinA1 (2 to 3-monthold male) mice were maintained by crossing Ai-IRKO mice with SerpinA1 +/- and IRf/f mice with SerpinA1 +/-, both of which were generated by breeding the offspring of Ai-IRKO mice with SPA1KO mice. For the induction of recombination, the mice were treated with 100 mg/kg tamoxifen (Sigma, T5648-1G) dissolved in 10% ethanol and 90% peanut oil (Sigma, P2144) by intraperitoneal injection five times over a six-day period starting at 2 months of age. In these experiments, tamoxifen was administered to all animal groups, including control mice, which carry *floxed alleles* but lack the adiponectin-CreER^T2 transgene[15].

We generated liver-specific SerpinA1-overexpressing transgenic (SPA1Tg) mice using cDNA encoding 3×Flag-tagged human *SERPINA1* cloned into an expression vector under the control of the mouse albumin promoter, as shown in Fig. 5a. The isolated and purified linearized transgene (3301 bp) was microinjected into mouse fertilized eggs (C57BL/6 J), and the transgenic offspring were screened by PCR-based genotyping using the following two sets of primers with identical results:

Forward ①: 5'-GCTCCATGCCCTAAAGAGAA-3'
Reverse ①: 5'-ATAGGCTGAAGGCGAACTCA-3'
Forward ②: 5'-CTACAAAGACCATGACGGTGA-3'
Reverse ②: 5'-GTGGTTCCCAACTCAGCAAC-3'.

For BAT-specific EphB2 knockdown experiments, 5 μg of siRNA (siControl: Accell Non-targeting Pool (Horizon Discovery/Dharmacon, D-001910-10-20), siEphB2: Accell Mouse Ephb2 (13844) siRNA – SMARTpool (Horizon Discovery/Dharmacon, E-050820-00-0020)) was directly injected into BAT of CD-fed male and female SPA1Tg mice using in vivo-jetPEI® (Polyplus, 101000040). Cold exposure experiments were performed 4 days after injection, and these mice were dissected 5 days after injection.

We generated whole-body SerpinA1 knockout (SPA1KO) mice by targeting all five *SERPINA1* paralogs using CRISPR–Cas9. SPA1KO mice were generated by introducing Cas9 protein (317−08441; Nippon Gene, Toyama, Japan), tracrRNA (GE-002; FASMAC, Kanagawa, Japan), synthetic crRNA (FASMAC), and ssODN into C57BL/6 J fertilized eggs using electroporation. To delete all five *SerpinA1* genes (*SerpinA1a-e*),

synthetic crRNAs were designed to direct GAGAGGTCCATGCTT-CACTG(AGG) in the 5′-neighboring region of SerpinA1b and AAACCTTTGAGGCAAACGCG(GGG) in the 3′-neighboring region of SerpinA1e. ssODN (5′-ATCCAGGAAAGTCTTCTGCACGCAACAGCGATG GCCTCAGGCGCATCGTAACCGTGCATGCGGGGGTGGGTGTGGGAGG GCTTCAGGTGGAGAACCTAC-3′) was used as a homologous recombination template.

The electroporation solutions consisted of 10 mM tracrRNA, 10 mM synthetic crRNA, 0.1 mg/mL Cas9 protein, and 1 mg/mL ssODN for SerpinA1(a-e) knockout in Opti-MEM I Reduced Serum Medium (31985062; Thermo Fisher Scientific). Electroporation was carried out using the Super Electroporator NEPA 21 (Nepa Gene, Chiba, Japan) on glass microslides with round wire electrodes and a 1.0 mm gap (45–0104; BTX, Holliston, MA). Square pulses were applied in 4 steps (1, three 3 mS poring pulses with 97 mS intervals at 30 V; 2, three 3 mS polarity-changed poring pulses with 97 mS intervals at 30 V; 3, five 50 mS transfer pulses with 50 mS intervals at 4 V with 40% decay of voltage per pulse; 4, five 50 mS polarity-changed transfer pulses with 50 mS intervals at 4 V with 40% decay of voltage per pulse). The resulting mice were genotyped by PCR using the following primers:

KO allele forward: 5′-TACATGCCCTGGACCCTCT-3′
KO allele reverse: 5′-GACACCAAGCTCAGTGCTCA-3′
WT allele forward: 5′-CAAACCTGGGGAGACTTTG-3′
WT allele reverse: 5′-AGAGTAGGGAACGTGATG-3′.

### Metabolic studies

An intraperitoneal glucose tolerance test (ipGTT) (2 mg/g body weight) was performed on mice after fasting for 6 h. Blood glucose levels were measured by collecting tail vein blood at 0, 15, 30, 60 and 120 min after glucose injection.

An intraperitoneal insulin tolerance test (ipITT) (1.0 U/kg body weight, Insulin Human R, Lilly) was performed on mice after fasting for 6 h. Blood glucose levels were measured by collecting tail vein blood at 0, 15, 30, 60 and 120 min after insulin injection.

Serum concentrations of mouse insulin (Alpco, 80-INSMSU-E01), mouse SerpinA1 (Abcam, ab205088) and human SerpinA1 (Immunology Consultants Laboratory, Inc., E-80A1T) were measured by ELISA.

Micro-CT imaging was performed using In-vivo Micro-CT scanners (ALOKA, LaTheta LCT-100).

For $O_2$ consumption metabolism measurement, mice were individually housed in climate-controlled rodent incubators (Muromachi Kikai, Tokyo, Japan, MK-5000RQ) at 23 °C on a 12-hour light-dark cycle. During this period, the mice had free access to food and water.

### Cold exposure

The mice were individually exposed to a 4 °C environment in climate-controlled rodent incubators (Muromachi Kikai, Tokyo, Japan, MK-5000RQ) for 3 h. During this period, the mice had no access to food or water. Rectal temperature and interscapular BAT temperature were measured after cold exposure (for 0, 30, 60, 90, 120, 150 or 180 min). A rectal probe (Physitemp Instruments, BAT-7001H) was used for rectal temperature measurement, and an implantable temperature transponder (BioMedic Data Systems, Inc., IPTT-300) was used for interscapular BAT temperature measurement. Interscapular BAT temperature was visualized using an infrared camera (Teledyne FLIR, FLIR E53 Advanced Thermal Imaging Infrared Camera).

### Immunohistochemistry

Tissues were fixed with neutral-buffered 10% formalin, embedded in paraffin, sectioned, and stained with hematoxylin and eosin (H&E) or immunostained with anti-UCP1 (Abcam, ab10983, 1:100) and anti-perilipin1 (Abcam, ab61682, 1:100) antibodies followed by incubation with the secondary antibody conjugated with Alexa 594 (Invitrogen, A32754, 1:500) and 488 (Invitrogen, A32814, 1:500).

### Adipocyte size analysis

Tissues were fixed with neutral-buffered 10% formalin, embedded in paraffin, sectioned, and stained with H&E. The adipocyte area was determined in auto mode (manual edition off) with Adiposoft image analysis software version 1.16. The conditions were a minimum diameter of 3 μm and a maximum diameter of 500 μm for CD-fed mice and a minimum diameter of 20 μm and a maximum diameter of 500 μm for HFD-fed mice.

### Quantification and statistical analysis

All data are presented as the mean ± SEM and were analyzed by two-tailed Student's $t$ test, one-way ANOVA post hoc Bonferroni test or two-tailed Student's $t$ test with Bonferroni's correction, as appropriate. Differences were considered significant if $p < 0.05$ according to two-tailed Student's $t$ test or one-way ANOVA post hoc Bonferroni test or if $p < 0.0167$ according to two-tailed Student's $t$ test with Bonferroni's correction. Statistics were carried out using GraphPad 7 or Excel version 2407. "$n$" indicates the number of animals per group or the number of independent experiments.

### Reporting summary

Further information on research design is available in the Nature Portfolio Reporting Summary linked to this article.

## Data availability

All data generated in this study is provided in the Supplementary Information/Source Data file. The raw mass spectrometry data (related Fig. 1b) have been deposited in the ProteomeXchange Consortium via the PRIDE partner repository under accession code PXD047144. The data are also available through the MassIVE repository (massive.ucsd.edu) with the dataset accession number MSV000093454. The raw mass spectrometry proteomics data (related Fig. 3i) have been deposited in the ProteomeXchange Consortium via the PRIDE partner repository under accession code PXD056002. Further information and requests for resources and reagents should be directed to and will be fulfilled by the lead contact, Masaji Sakaguchi (msakaguchi@kumamoto-u.ac.jp). Source data are provided with this paper.

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

## Acknowledgements

We would like to thank members of the Center for Animal Resources and Development at Kumamoto University for their important contributions to the experiments. This work was supported by the Japan Society for the Promotion of Science KAKENHI (grant numbers JP 18K16208, JP 16H06276 (AdAMS) and JP 21K08532), Grants for Young Researchers from the Japan Association for Diabetes Education and Care (2021), the Kumamoto University Hospital Research Revitalization Project (2023), a grant from the Manpei Suzuki Diabetes Foundation (2021), a grant from the Japan Society for the Study of Obesity and Novo Nordisk Pharma Ltd. (2022), Astellas Foundation for Research on Metabolic Disorders (2022) and a grant from the Center for Metabolic Regulation of Healthy Aging (2021) to M.S. This research is also supported by AMED under Grant Number JP24gm6910015 to M.S. This work was supported by a grant from the Japan Foundation for Applied Enzymology (2023) to S.O. R.N.K. acknowledges support from the NIH grant R01 DK067536.

## Author contributions

S.O. designed and performed the experiments, analyzed the data, and wrote the paper. Y.O., Y. T., M.I., T.K., B.B.B., W.J.Q., Y.H.T., R.N.K., N.K., C.R.K. and E.A. helped to perform the experiments. N.T. generated the SPA1Tg mice. K.A. designed and generated the SPA1KO mice. M.S. supervised the project, designed and performed the experiments, analyzed the data, and wrote the paper.

## Competing interests

The authors declare no competing interests.
