## [Transparent Peer Review file · Nature Communications]

Hepatic SerpinA1 Improves Energy and Glucose Metabolism through Regulation of Preadipocyte Proliferation and UCP1 Expression

Corresponding Author: Dr Masaji Sakaguchi

Version 0:

Reviewer comments:

Reviewer #1

(Remarks to the Author)

Okagawa et al. identified a serine protease inhibitor A1 (SerpinA1) in the liver that could activate brown fat and promote browning of white fat as well as increasing the expression of uncoupling protein 1 (UCP1) to promote mitochondrial activation in mature white and brown adipocytes. Liver-specific SerpinA1 transgenic mice exhibit increased browning of adipose tissues, leading to improved glucose tolerance. Conversely, SerpinA1 knockout (SPA1KO) mice show adipocyte mitochondrial dysfunction, impaired glucose tolerance and reduced systemic insulin resistance. Mechanistically, SerpinA1 binds to the EphrinB2 receptor, to control p38 phosphorylation and subsequently regulating UCP1 expression.

This study is novel as no studies to date have examined the role of SerpinA1 on adipocyte function and whole-body metabolism. The methodology used is sound and enough detail is provided in the methods for the work to be reproduced.

Additional questions:

1. SerpinA1 has multiple paralogues, did the authors measure each one separately in other tissues besides the liver?
2. In Figure 1D, did the authors check SerpinA1 expression in WAT?
3. No rationale for the use of male mice. Are there any sex specific differences between males and females?
4. Body weights in SPA1Tg mice under chow fed conditions were unchanged despite a slight reduction in iWAT, so are there any changes in lean mass?
5. Any changes in insulin signaling in the liver?
6. Was pancreatic function measured? How does SerpinA1 act on the pancreas to reduce insulin secretion?
7. How does SerpinA1 increase mitochondrial function? Through which pathway?

Reviewer #2

(Remarks to the Author)

This study aims to further decipher the metabolic functions of serpin-A1 / alpha1 antitrypsin. Using mechanistic investigations in different mice models (insulin receptor/IGF1R KO with lipoatrophy, adipocyte-specific IR-only KO mice, liver-specific overexpression or knockout of serpin-A1, whole body serpin-A1 KO mice) the authors show that serpinA1 promotes preadipocyte proliferation and browning of adipose tissue, increased energy expenditure, reduced adiposity and improved glucose tolerance and insulin sensitivity.

The authors identify EphB2 as a partner of serpinA1 in adipocytes, which is necessary for the effects of serpinA1 on proliferation of brown adipocytes and for the induction of UCP1 expression upon serpinA1 stimulation.

The translational impact of the results in humans would be important to study further.

Major comments

The authors investigate the level of serpinA1 in adipocyte-specific IR-only KO mice (fig 1c and line 137). Could they explain

what is the importance of this finding : do these mice also recover from lipodystrophy ? Were proteomics analysis performed in these mice? What were the results in IGFR1 only KO mice?

The description of the results in the discussion section could be shortened. Please comment further on the metabolic phenotype in patients with alpha 1 antitrypsin deficiency, or treated by inhibitors of alpha 1 antitrypsin, and conversely on the liver and lung phenotype in serpinA1 KO mice.

Could the authors provide any result showing the involvement of EphB2 in the effects of serpin-A1 in vivo ?

Minor comments

Introduction

Line 66 the dysregulated balance between WAT and BAT is probably not the main driver of obesity-related complications in humans, could the authors modify this sentence?

Line 84 "Thus activation of BAT and browning of WAT tend to protect against body fat accumulation and its comorbidities". Could the authors provide references, in addition to correlations cited above, to support that browning of fat was shown to protect against comorbidities associated with obesity in humans?

Fig 2 legend : please specify the model of white adipocytes used in these experiments

Line 159 the effect of serpin A1 on oil red O staining seems different in figure 4 and suppl figure 2, could you explain these discrepancies ?

Line 172-174 do the authors could provide other results showing the conversion from white to beige adipocytes ? could they show other markers of beige adipocytes in these cells ?

Line 196-198 could the authors explain the link between ERK1/2 signal and beta3 adrenergic receptor-mediated stimulation of adipocyte browning?

Figure 3 and line 206-208 : Since activation of Erk1/2 signal pathways is not specific to serpinA1 or EphB2 stimulation, could the authors show the effects of serpinA1 in EphB2 positive or negative cells on other markers of cell proliferation? The same is true for p38 activation as a determinant of increased UCP1 expression (Figure 4 and line 229-230).

Figure 5 and related text: please specify whether SPA1Tg mice were fed a chow diet for these experiments? if this is confirmed, since their total weight is not modified as compared to controls, did they display an increased lean mass (fig 5a vs suppl fig 6a), which could explain the metabolic improvements ?

Figure 6 a-e: Fed a chow diet, SPA1Tg mice show improved glucose tolerance without increased insulin sensitivity : is glucose-induced insulin secretion increased in these mice ?

Fig 6f-g: is the decreased body weight of HFD-SPA1Tg mice due to a decreased amount of adipose tissue?

Reviewer #3

(Remarks to the Author)

In this manuscript, the authors studied the role of SerpinA1 in regulating energy and glucose metabolism. They found that SerpinA1 promoted preadipocyte proliferation and UCP1 expression which improved mitochondrial function in mature adipocytes. Serpin A1 interacted with EphB2 and regulated its downstream signaling in adipocytes. Overexpression of SerpinA1 in mice increased mitochondrial function, energy expenditure, and glucose tolerance. Knockout of SerpinA1 in mice decreased mitochondrial function, glucose tolerance and increased obesity. There are some novelties and interesting findings to study the mechanisms of SerpinA1 in regulating adipocytes. However, some concerns need to be addressed to make the conclusions more solid.

Specific comments:

1. SerpinA1 was upregulated in Ai-DKO mice after tamoxifen withdraw through proteomics. It will be more interesting and convincing if the authors show SerpinA1 affects adipocyte recovery in Ai-DKO mice.
2. Figures 1 and 2 showed that white or brown preadipocytes treated with SerpinA1 increased the proliferation and improved mitochondrial function. What's the blood concentration of SerpinA1 in Ai-DKO mice? The authors used different doses of recombinant SerpinA1 to treat white or brown preadipocytes or for different assays (proliferation vs seahorse). What're the rationales that the doses were chosen for different cells or assays? Are these doses physiological relevant?
3. Figure 3 showed SerpinA1 interacted with EphB2 exogenously in adipocytes. Is there endogenous interaction of these two proteins? How is in vivo? The quality of some blots needs to be improved, such as 3k.
4. Figures 5 and 6 showed that SerpinA1 Tg mice had similar adipose tissue weight but smaller adipocyte sizes in chow- or HFD-fed mice. What're the explanations for these results? Echo-MRI for body composition in mice would be more accurate. Is there food intake difference in Tg mice? What're SerpinA1 blood concentrations in Tg mice with chow or HFD? What're the rationales to use different ages of mice in different assays? For examples, Fig 5a-5h used 12-week-old, Fig 5i 8-week-old, Fig 5k-l 18-week-old. The authors concluded that SerpinA1 Tg increased energy expenditure. The data is not

convincing enough. It's better to include indirect calorimetry assay to the study.

5. Figure 7 showed SerpinA1 KO mice had opposite effects as Tg mice. Similar concerns as above.

6. Male mice were used in whole study. Are there any gender differences? It's better to show both genders.

7. While the study focused on adipose tissues in mice, did the authors notice any abnormalities in Tg or KO mouse livers?

Version 1:

Reviewer comments:

Reviewer #1

(Remarks to the Author)

The authors have adequately addressed the concerns/comments of the revised manuscript.

Reviewer #2

(Remarks to the Author)

The authors have carefully responded to all the points I previously raised, I do not have any other comment

Reviewer #3

(Remarks to the Author)

In this revised manuscript, the authors performed extra experiments to address the comments raised by the reviewers. The manuscript is significantly improved, and the study is technically sound. This reviewer has no further comments.

The Point-by Point Responses

Manuscript: Hepatic SerpinA1 Improves Energy and Glucose Metabolism. through. Regulation of Preadipocyte Proliferation and UCP1 Expression

Manuscript #: NCOMMS-24-01048-T

Corresponding Author: Masaji Sakaguchi

REVIEWER COMMENTS

Reviewer #1 (Remarks to the Author):

Okagawa et al. identified a serine protease inhibitor A1 (SerpinA1) in the liver that could activate brown fat and promote browning of white fat as well as increasing the expression of uncoupling protein 1 (UCP1) to promote mitochondrial activation in mature white and brown adipocytes. Liver-specific SerpinA1 transgenic mice exhibit increased browning of adipose tissues, leading to improved glucose tolerance. Conversely, SerpinA1 knockout (SPA1KO) mice show adipocyte mitochondrial dysfunction, impaired glucose tolerance and reduced systemic insulin resistance. Mechanistically, SerpinA1 binds to the EphrinB2 receptor, to control p38 phosphorylation and subsequently regulating UCP1 expression.

This study is novel as no studies to date have examined the role of SerpinA1 on adipocyte function and whole-body metabolism. The methodology used is sound and enough detail is provided in the methods for the work to be reproduced.

---We appreciate the reviewer's evaluation of our manuscript and their helpful advice. Owing to your comments, we believe our manuscript has improved.

Additional questions:

1. SerpinA1 has multiple paralogues, did the authors measure each one separately in other tissues besides the liver?

--- Thank you for your question. We analyzed the expression of each mouse SerpinA1 paralog (SerpinA1a-e) with specific primers in various organs, including the liver. Our results show that all SerpinA1 paralogs are similarly expressed in the liver, while almost little

expression was observed in other organs. We have added these data in the text (p. 8, lines 152-157) with Figure 1d and Supplementary Figure 1a.

2. In Figure 1D, did the authors check SerpinA1 expression in WAT?

--- We analyzed SerpinA1 expression in various types of white adipose tissue (WAT), including subcutaneous, perigonadal, and retroperitoneal WAT. Our results showed that SerpinA1 expression was undetectable under our standard PCR analysis in any of these WAT types. These data have been described in the text (p. 8, lines 152-157) with Figure 1d and Supplementary Figure 1a.

3. No rationale for the use of male mice. Are there any sex specific differences between males and females?

--- Thank you for bring up this point. In fact, we have studied both sexes and found no significant differences in the effect of SerpinA1 between males and females, although female SPA1Tg mice showed more fat mass reduction than males, particularly in perigonadal WAT. We have added these data in the text (p. 13, lines 265-269) with Figure 5b and Supplementary Figure 5h. We have used male and female mice for comparison in most of experiments in this manuscript as lines 153, 265, 266, 267, 270, 273, 274, 283, 286, 292, 300, 303, 314, 315, 316, 318, 322, 355, 362 and 368.

4. Body weights in SPA1Tg mice under chow fed conditions were unchanged despite a slight reduction in iWAT, so are there any changes in lean mass?

--- In response to your comment, we performed CT scans on SPA1Tg mice to measure lean mass. The results show a slight increase in the proportion of lean mass, but this change is not statistically significant (Supplementary Figure 5j). Overall body weight was unchanged, possibly due to a slight increase in lean mass under the chow diet condition. Under high-fat diet feeding conditions, the mice displayed a more significant decrease in body weight, which was parallel with the decreased weight of adipose tissues. We explained this in the Results section (p. 13, lines 272-276) and the Discussion section (p. 25, lines 518-525).

5. Any changes in insulin signaling in the liver?

--- To address your question, we administered insulin via the inferior vena cava and analyzed

insulin signaling in the liver with the IR, Akt, and ERK phosphorylation using western blotting (Supplementary Figures 7h, i). The results showed no differences in insulin signaling between SPA1KO mice and control mice. We explain this in the text (p.17, lines 359-361).

6. Was pancreatic function measured? How does SerpinA1 act on the pancreas to reduce insulin secretion?

--- In response to the reviewers' concerns, we have now measured glucose-stimulated insulin secretion in 12-week-old SPA1Tg mice. The SPA1Tg mice displayed lower glucose-stimulated insulin secretion, yet their blood glucose levels were controlled. Insulin tolerance testing performed on the same 12-week-old SPA1Tg mice showed an improvement in insulin sensitivity compared to control mice. These findings suggest that the reduction in insulin secretion is due to improved insulin sensitivity secondary to activation of white and brown adipose tissues. These results have been added to Figures 6b-e and Supplementary Figures 6f-k and explained in the text (p.15, line 317 to p.16, line 323).

7. How does SerpinA1 increase mitochondrial function? Through which pathway?

--- The activation of brown adipocytes results from mitochondrial activation with a proton transporter UCP1 to cause a leak of protons from the intermembrane space of mitochondria. Seahorse Analysis revealed that SerpinA1 increased the maximal oxygen consumption rate along with an elevation in UCP1 expression in brown adipocytes. To determine if SerpinA1 regulates mitochondrial function in a UCP1-dependent manner, we tested with UCP1 knockdown (UCP1-KD) brown adipocytes. SerpinA1 treatment did not result in significant changes in oxygen consumption rate in UCP1-KD mature brown adipocytes, suggesting that the observed changes are UCP1-dependent. We have now included these data in the text (p.9, lines 178-180) in the Supplementary Figures 2b-d.

Reviewer #2 (Remarks to the Author):

This study aims to further decipher the metabolic functions of serpin-A1 / alpha1 antitrypsin. Using mechanistic investigations in different mice models (insulin receptor/IGF1R KO with lipoatrophy, adipocyte-specific IR-only KO mice, liver-specific overexpression or knockout of serpin-A1, whole body serpin-A1 KO mice) the authors show that serpinA1 promotes preadipocyte proliferation and browning of adipose tissue, increased energy expenditure, reduced adiposity and improved glucose tolerance and insulin sensitivity.

The authors identify EphB2 as a partner of serpinA1 in adipocytes, which is necessary for the effects of serpinA1 on proliferation of brown adipocytes and for the induction of UCP1 expression upon serpinA1 stimulation.

The translational impact of the results in humans would be important to study further.

---We appreciate your thorough review and helpful advice. Owing to your comments, our manuscript has improved (see detailed response below). We agree that studying the translational impact of these results in humans is crucial and plan to address this in future research.

Major comments

The authors investigate the level of serpinA1 in adipocyte-specific IR-only KO mice (fig 1c and line 137). Could they explain what is the importance of this finding: do these mice also recover from lipodystrophy? Were proteomics analysis performed in these mice? What were the results in IGFR1 only KO mice?

---Thank you for your comments. We have added more explanation in the text. As previously reported by us in Cell Metabolism 2017, IRKO and DKO mice exhibit severe lipodystrophy, impaired glucose tolerance, and insulin resistance. Interestingly, they are able to recover from lipodystrophy and metabolic syndrome when the tamoxifen induction is stopped. Therefore, we can analyze the molecular changes in IRKO and DKO mice during the induction of lipodystrophy, in the lipodystrophic state and during its recovery. Although full proteomics analyses were conducted only in Control and DKO mice, we compared the concentration of SerpinA1 in the serum of Control, IGF1RKO, IRKO, and DKO mice using ELISA. The results showed an increase in SerpinA1 concentration in IRKO and DKO mice, whereas there was no significant difference between IGF1RKO and Control mice. These data have been added to Figure 1c and explained in the text (p. 7, line 143 to p. 8, line 148).

The description of the results in the discussion section could be shortened. Please comment further on the metabolic phenotype in patients with alpha 1 antitrypsin deficiency, or treated by inhibitors of alpha 1 antitrypsin, and conversely on the liver and lung phenotype in serpinA1 KO mice.

--- Thank you for your insightful suggestions. We have shortened the description in the discussion section. We now describe the metabolic phenotype in patients with alpha-1 antitrypsin deficiency and how it differs from that in SerpinA1 KO mice. The alpha 1

antitrypsin deficiency (A1ATD; SerpinA1) patients with the mutated gene showed liver abnormalities, such as cirrhosis, caused by the accumulation of the mutated alpha 1 antitrypsin protein in hepatocytes. The burden of the misfolded, aggregated A1AT protein precipitates the development of liver disease. On the other hand, the lack of A1AT in the systemic circulation, exacerbated by factors such as smoking, increases susceptibility to lung injury, early-onset lung emphysema, and chronic obstructive pulmonary disease (COPD). Consequently, individuals who do not produce A1AT (null/null) are highly susceptible to lung injury but typically present with normal livers. Indeed, fazirsiran, an RNAi therapeutic candidate designed to reduce the production of the mutated alpha one antitrypsin protein, has been reported to lead to liver repair (Pavel S. et al., NEJM 2022).

In our study, SerpinA1 KO mice showed no significant abnormalities in liver morphology, liver weight, histological analysis, or gene expression (Supplementary Figures 7e-g). SerpinA1 KO mice showed no morphological or histological abnormalities in the lungs (Supplementary Figures 7c-d). Previous reports have indicated that A1AT KO mice will develop pulmonary emphysema, but only at an advanced age (>35 weeks), consistent with the absence of abnormalities in the young adult SerpinA1 KO mice used in our study. In the future, we aim to analyze the relationship between expression levels of SerpinA1 and metabolic conditions, such as diabetes and adiposity, not only in patients with SerpinA1 gene mutations, but also through quantitative intervention studies on alpha 1 antitrypsin (SerpinA1) level. We discussed this in the Discussion section (p.21, line 435 to p. 22, line 445).

Could the authors provide any result showing the involvement of EphB2 in the effects of serpin-A1 *in vivo* ?

---Thank you for your question. To investigate the involvement of EphB2 in the effect of SerpinA1 *in vivo*, we have now employed the small interfering RNA (siRNA) administration technique, as described in Jeon et al., Nature Communications 2024. Knockdown of EphB2 in the brown adipose tissue blocked the increase of UCP1 expression and thermogenesis under cold stimulation of SPA1Tg mice. These results suggest that EphB2 is involved in the SerpinA1-induced activation of brown adipose tissues *in vivo* (p. 14, line 297 to p. 15, line 307) in Supplementary Figures 5r-v.

Minor comments

Introduction

Line 66 the dysregulated balance between WAT and BAT is probably not the main driver of

obesity-related complications in humans, could the authors modify this sentence?

--- We have modified the sentence to reflect the insights from Becher et al. (Nature Medicine, 2021), acknowledging that these perturbations are associated with individuals with lower amounts of BAT. This updated perspective has been incorporated into the manuscript (p. 4, lines 66-67).

Line 84 "Thus activation of BAT and browning of WAT tend to protect against body fat accumulation and its comorbidities".

Could the authors provide references, in addition to correlations cited above, to support that browning of fat was shown to protect against comorbidities associated with obesity in humans?

---Thank you for your suggestion. In response to your request, we have revised the text to reflect the current state of research more accurately. The revised statement now reads: "Thus, activation of BAT and browning of WAT appear to protect against body fat accumulation and its comorbidities." with the references (Becher et al. 2021) in the text (p. 5, lines 85-91).

Fig 2 legend : please specify the model of white adipocytes used in these experiments

---Thank you for your suggestion. The white adipocytes used in these experiments are referred to as A41 hWAT-SVF cells and were provided by the co-author Dr. Yu-Hua Tseng. These cells are white preadipocytes collected from a human subject and immortalized by hTERT overexpression. They are currently stored in the ATCC bank under the designation CRL-3386. We have added this information to the Materials and Methods section (p. 40, lines 846-847).

Line 159 the effect of serpin A1 on oil red O staining seems different in figure 4 and suppl figure 2, could you explain these discrepancies?

--- Thank you for your careful review. The experiments shown in Figure 4 and Supplementary Figure 2 were conducted with different cohorts, and the sizes of the cell culture wells were different. This resulted in differences in cell density, which may have caused the control cells to appear different. To avoid misunderstandings, we have now replaced the figure with data performed under from the same conditions in Supplementary Figure 2.

Line 172-174 do the authors could provide other results showing the conversion from white to beige adipocytes ? could they show other markers of beige adipocytes in these cells ?

--- We measured the beige markers DiO₂, Pat2, CD137, CD40, CITED1, and Sp100, which were significantly increased in white adipocytes stimulated with SerpinA1. In addition to activating UCP1 and mitochondrial function, these results are consistent with the conversion to beige adipocytes. We have included these results in Supplementary Figure 2g provided explanations in the text (p. 9, lines 182-184).

Line 196-198 could the authors explain the link between ERK1/2 signal and beta3 adrenergic receptor-mediated stimulation of adipocyte browning?

---Thank you for your comment. In response to your query, we have clarified the connection between ERK1/2 signaling and beta3 adrenergic receptor-mediated stimulation of adipocyte browning. To avoid misunderstanding, we have revised the text to emphasize the role of ERK1/2 in cSCC cell proliferation and its association with brown preadipocyte proliferation without implying a direct link to browning mediated by beta3 adrenergic receptors. The corrected sentence is as follows: "EphB2 promotes cutaneous squamous cell carcinoma (cSCC) cell proliferation via the Erk1/2 signaling pathway. The Erk1/2 signaling pathway is also linked to brown preadipocyte proliferation." (inserted on p. 10, line 211 to p. 11, line 214).

Figure 3 and line 206-208 : Since activation of Erk1/2 signal pathways is not specific to serpinA1 or EphB2 stimulation, could the authors show the effects of serpinA1 in EphB2 positive or negative cells on other markers of cell proliferation? The same is true for p38 activation as a determinant of increased UCP1 expression (Figure 4 and line 229-230).

--- We agree with the reviewer and acknowledge the need to measure other markers besides Erk1/2 in preadipocytes. We have now measured FAK phosphorylation, which is well-known to be a downstream signaling pathway of EphB2 (Moeller ML et al., 2006), and found that SerpinA1-treated preadipocytes exhibited an increase in FAK phosphorylation. Additionally, we measured FAK phosphorylation in mature adipocytes and found that FAK phosphorylation was significantly increased in SerpinA1-treated mature adipocytes. Furthermore, in EphB2KO cells, we found that these increases in FAK phosphorylation were lost. We have explained these data in the text (p. 11 line 222-226 and p.12 line 247-251) with Supplementary Figures 3c, d and 4a, b.

Figure 5 and related text: please specify whether SPA1Tg mice were fed a chow diet for these experiments? if this is confirmed, since their total weight is not modified as compared to controls, did they display an increased lean mass (fig 5a vs suppl fig 6a), which could explain the metabolic improvements ?

--- Thank you for your comment. We have now performed CT scans on SPA1Tg mice to measure lean mass. Additionally, in Figure 5 and the related text, we have specified that these experiments were conducted with SPA1Tg mice fed a chow diet. The results showed a slight increase in the proportion of lean mass, but this change was not statistically significant (Supplementary Figure 5j). The overall body weight seemed unchanged, possibly due to only a slight increase in lean mass under the chow diet condition. Alternatively, on a chow diet, the reduction in fat tissue weight was slight and a less noticeable difference in body weight compared to controls since the proportion of fat mass was lower. High-fat diet feeding resulted in a more significant difference in body weight, which was parallel with the decreased weights of adipose tissues. We explained this in the Results section (p. 13, lines 272-276) and the Discussion section (p. 25, lines 518-525).

Figure 6 a-e: Fed a chow diet, SPA1Tg mice show improved glucose tolerance without increased insulin sensitivity : is glucose-induced insulin secretion increased in these mice ?

---Thank you for your comment. We have now measured pancreatic function by assessing glucose-stimulated insulin secretion in 12-week-old SPA1Tg mice. The SPA1Tg mice displayed lower insulin secretion, yet their blood glucose levels were controlled. An insulin tolerance test was performed on the same 12-week-old mice. SPA1Tg mice exhibited improved insulin sensitivity compared to control mice, although we could not clearly show significant differences in 8-week-old mice compared with the control. These findings suggest reduced insulin secretion may be attributed to improved insulin sensitivity. We have described the results in the text (p.15, line 317 to p.16, line 323) and included the additional data in Figures 6b-e and Supplementary Figures 6f-k.

Fig 6f-g: is the decreased body weight of HFD-SPA1Tg mice due to a decreased amount of adipose tissue?

--- Thank you for your comment. As the reviewer suggested, we have now measured adipose tissue mass in HFD-fed SPA1Tg mice. We found that HFD-fed SPA1Tg mice showed significant decreases in both iWAT and pgWAT. These results indicate that decreased

adipose tissue contributes to the decreased body weight observed in HFD-SPA1Tg mice. We have added these results in Supplementary Figure 6l and provided explanations in the text (p. 16, lines 327-329).

Reviewer #3 (Remarks to the Author):

In this manuscript, the authors studied the role of SerpinA1 in regulating energy and glucose metabolism. They found that SerpinA1 promoted preadipocyte proliferation and UCP1 expression which improved mitochondrial function in mature adipocytes. Serpin A1 interacted with EphB2 and regulated its downstream signaling in adipocytes. Overexpression of SerpinA1 in mice increased mitochondrial function, energy expenditure, and glucose tolerance. Knockout of SerpinA1 in mice decreased mitochondrial function, glucose tolerance and increased obesity. There are some novelties and interesting findings to study the mechanisms of SerpinA1 in regulating adipocytes. However, some concerns need to be addressed to make the conclusions more solid.

--- Thank you very much for your overall evaluation of our manuscript. Your constructive comments have improved our revised manuscript significantly.

Specific comments:

1. SerpinA1 was upregulated in Ai-DKO mice after tamoxifen withdraw through proteomics. It will be more interesting and convincing if the authors show SerpinA1 affects adipocyte recovery in Ai-DKO mice.

---Thank you for your insightful suggestion. As previously reported (Sakaguchi et al., Cell Metabolism 2017), we observed adipose recovery in IR/IGF1R DKO mice and IRKO mice. We confirmed that SerpinA1 levels are elevated in both DKO and IRKO mice. To investigate the effect of SerpinA1 on adipose recovery, we have now crossed IRKO mice with SerpinA1 KO mice and conducted a comparative analysis. Our results show that the absence of SerpinA1 leads to reduced recovery of white and brown adipose tissues and a decreased thermogenic function in brown adipose tissue. We have included these data in Figures 1c, 7j-k and Supplementary Figures 7r-t and provided explanations in the text (p. 7, line 143 to p. 8, line 148 and p. 18, line 376 to p. 19, line 392). These findings demonstrate the role of SerpinA1 in adipose tissue recovery.

2. Figures 1 and 2 showed that white or brown preadipocytes treated with SerpinA1

increased the proliferation and improved mitochondrial function. What's the blood concentration of SerpinA1 in Ai-DKO mice? The authors used different doses of recombinant SerpinA1 to treat white or brown preadipocytes or for different assays (proliferation vs Seahorse). What're the rationales that the doses were chosen for different cells or assays? Are these doses physiological relevant?

--- Thank you for your comment. As the reviewer pointed out, excessively high concentrations of SerpinA1 may not be physiologically relevant. Therefore, we have now used concentrations based on the serum levels detected *in vivo*. In wild-type mice, the serum concentration of SerpinA1 is approximately 300 mg/dL, which is increased to around 450 mg/dL in Ai-DKO mice. These data have been added to Figure 1c. As mentioned in Figure legends, we used doses between 50 and 500 µg/mL of SerpinA1 to create levels within the physiological range for *in vitro* experiments. SerpinA1 was freshly prepared and used immediately for each experiment cohort to ensure consistency in experimental conditions. We explained the dose of the physiological relevance in the Methods section (p. 41, lines 868-870).

3. Figure 3 showed SerpinA1 interacted with EphB2 exogenously in adipocytes. Is there endogenous interaction of these two proteins? How is *in vivo*? The quality of some blots needs to be improved, such as 3k.

---As the reviewer suggested, we analyzed the endogenous interaction between SerpinA1 and EphB2 using immunoprecipitation with lysates from the tissues. Our result confirmed that SerpinA1 interacts with EphB2 endogenously. We have updated this data (Supplementary Figure 3b) in the revised manuscript (p. 10, lines 201-203).

4. Figures 5 and 6 showed that SerpinA1 Tg mice had similar adipose tissue weight but smaller adipocyte sizes in chow- or HFD-fed mice. What're the explanations for these results? Echo-MRI for body composition in mice would be more accurate.

---Thank you for your comment. SerpinA1Tg showed a slight decrease in adipose pad weights when on chow diet. In comparing adipocyte sizes, SerpinA1Tg showed a decrease in larger adipocytes (40-50 µm, 50-60 µm and above 60 µm) compared to control mice. However, the proportion of such larger adipocytes was relatively small even in control mice under a chow diet, this decrease might not have a significant impact on adipose tissue weight. Following the reviewer's suggestion, we have now performed a CT scan analysis to measure lean mass. The results show a slight increase in the proportion of the lean mass, but this

change was not significant. And fat mass is slightly reduced; this difference was statistically significant in both males and females, contributing to a less pronounced change in total fat weight (Supplementary Figure 5j). It is difficult to measure the change in fat mass weight compared with controls since the proportion of fat mass is low on chow diet, and the reduction in fat tissue weight is relatively small on a chow diet. Therefore, we have now measured fat weights under high-fat diet (HFD) conditions for SerpinA1Tg mice and observed significant reductions in both iWAT and pgWAT, which were in parallel with the decrease in body weights in both females and males. We have added these new data to Supplementary Figure 6l to provide a more comprehensive view of the fat mass differences under both diet conditions. We described these results in the Discussion section (p. 25, lines 518-525).

Is there food intake difference in Tg mice?

---Regarding food intake, we did not observe any differences between Tg and control mice (Supplementary Figure 6b), suggesting that the differences in fat weight are not due to food intake.

What're SerpinA1 blood concentrations in Tg mice with chow or HFD?

---We used the human SerpinA1 gene for our Tg mice. We measured endogenous SerpinA1 and human SerpinA1 using ELISA in CD-fed Tg mice. WT mice had SerpinA1 levels around 300 mg/dL, while Tg mice had similar levels of mouse-derived SerpinA1 and additional human-derived SerpinA1 at approximately 400 mg/dL (Supplementary Figure 5d).

What're the rationales to use different ages of mice in different assays? For examples, Fig 5a-5h used 12-week-old, Fig 5i 8-week-old, Fig 5k-l 18-week-old.

--- We have updated the data by consistently using 12-week-old mice across all experiments in Figure 5.

The authors concluded that SerpinA1 Tg increased energy expenditure. The data is not convincing enough. It's better to include indirect calorimetry assay to the study.

--- We acknowledge the suggestion and have now incorporated indirect calorimetry assays into our study. The results from these assays confirm that energy expenditure is increased in SerpinA1Tg mice. The relevant data and explanations are now included in the revised manuscript in Supplementary Figures 5p, q.

5. Figure 7 showed SerpinA1 KO mice had opposite effects as Tg mice. Similar concerns as above.

--- Thank you for your comment. We have further investigated the SerpinA1 KO mice. Specifically, we observed that the KO mice have larger adipocyte sizes, as the reviewer suggested, prompting us to measure the fat weights in these mice. Under a chow diet, we found significant increases in the fat weights of both iWAT and pgWAT, as well as under HFD conditions. These new measurements have been included in Supplementary Figure 7I to address the observed differences.

6. Male mice were used in whole study. Are there any gender differences? It's better to show both genders.

---- Thank you for the comment. We have now examined female mice and found no significant difference in the effect of SerpinA1 on brown adipocytes between males and females. However, female SPA1Tg mice showed more reduction in fat mass, particularly in pgWAT. We introduced these data in the text (p. 13, lines 265-269) with Figure 5b and Supplementary Figure 5h. We have used both male and female mice for comparison in most of experiments in this manuscript as lines 153, 265, 266, 267, 270, 273, 274, 283, 286, 292, 300, 303, 314, 315, 316, 318, 322, 355, 362 and 368.

7. While the study focused on adipose tissues in mice, did the authors notice any abnormalities in Tg or KO mouse livers?

--- Thank you for your question. In SerpinA1 KO mice, we did not observe any abnormalities in the liver weight, histological analysis, or gene expression analysis (Supplementary Figures 7e-g). We further analyzed insulin signaling in the liver via western blotting for pIR, pAkt, and pERK after insulin administration and also found no differences between SerpinA1KO and control mice (Supplementary Figures 7h,i). In patients with alpha 1 antitrypsin deficiency (A1ATD), liver abnormalities such as cirrhosis are primarily caused by the accumulation of the mutated alpha one antitrypsin protein in hepatocytes rather than by a reduction in the levels of alpha 1 antitrypsin. The burden of the misfolded, aggregated A1AT protein precipitates the development of liver disease. This might explain the absence of significant phenotypes in the KO mice. We have described this in the Discussion section (p.21, line 435 to p. 22, line 445) and added the data in Supplementary Figures 7e-i.